The EMBO Journal (2013) 32, 1250–1264
www.embojournal.org

# Distinct roles for Sir2 and RNAi in centromeric heterochromatin nucleation, spreading and maintenance

**Alessia Buscaino[1], Erwan Lejeune[1], Pauline Audergon, Georgina Hamilton, Alison Pidoux and Robin C Allshire***

Wellcome Trust Centre for Cell Biology, Institute of Cell Biology, School of Biological Sciences, The University of Edinburgh, Edinburgh, UK

**Epigenetically regulated heterochromatin domains govern essential cellular activities. A key feature of heterochromatin domains is the presence of hypoacetylated nucleosomes, which are methylated on lysine 9 of histone H3 (H3K9me). Here, we investigate the requirements for establishment, spreading and maintenance of heterochromatin using fission yeast centromeres as a paradigm. We show that establishment of heterochromatin on centromeric repeats is initiated at modular 'nucleation sites' by RNA interference (RNAi), ensuring the mitotic stability of centromere-bearing minichromosomes. We demonstrate that the histone deacetylases Sir2 and Clr3 and the chromodomain protein Swi6[HP1] are required for H3K9me spreading from nucleation sites, thus allowing formation of extended heterochromatin domains. We discovered that RNAi and Sir2 along with Swi6[HP1] operate in two independent pathways to maintain heterochromatin. Finally, we demonstrate that tethering of Sir2 is pivotal to the maintenance of heterochromatin at an ectopic locus in the absence of RNAi. These analyses reveal that Sir2, together with RNAi, are sufficient to ensure heterochromatin integrity and provide evidence for sequential establishment, spreading and maintenance steps in the assembly of centromeric heterochromatin.**

*The EMBO Journal* (2013) **32**, 1250–1264. doi:10.1038/emboj.2013.72; Published online 9 April 2013
*Subject Categories:* chromatin & transcription
*Keywords*: centromeres; epigenetics; HDACs; heterochromatin; RNAi

## Introduction

Chromatin assembly controls vital cellular activities in eukaryotes. Beyond creating modular DNA–protein scaffolds, the formation of chromatin domains is essential for accurate dosage compensation, lineage differentiation, chromosome compaction and epigenetic imprinting. Assembly of chromatin

*Corresponding author. Wellcome Trust Centre for Cell Biology, Institute of Cell Biology, School of Biological Sciences, The University of Edinburgh, Edinburgh EH9 3JR, Scotland, UK. Tel.: + 44 131 650 7117; E-mail: robin.allshire@ed.ac.uk
[1]These authors contributed equally to this work.

domains is thought to be a three-step process: establishment, spreading and maintenance (Rusche *et al*, 2003). However, the molecular mechanisms underlying these distinct stages remain to be determined. During establishment, naive chromatin acquires a specific epigenetic signature, characterized by particular histone post-translational modifications. This relies on inducers that trigger an altered chromatin state at specific locations, termed nucleation sites. Once the initial chromatin modification is established, it can then spread *in cis* over several kilobases of DNA, irrespective of its sequence. Additional factors may be required for the maintenance of these chromatin domains in the absence of the inducer (Berger *et al*, 2009). Heterochromatin domains inhibit gene expression and consequently tend to be gene-poor, but they also regulate key cellular processes, including recombination, DNA repair and chromosome segregation (Grewal, 2010). In most eukaryotes, large blocks of heterochromatin are found at centromeres (Buscaino *et al*, 2010). At these regions, histones are generally hypoacetylated and specifically methylated on H3K9 (H3K9me). H3K9me creates a binding site for chromodomain proteins, which complete the assembly of transcriptionally repressive chromatin (Rea *et al*, 2000; Nakayama *et al*, 2001; Sadaie *et al*, 2004). Heterochromatin integrity at centromeres can be monitored by the transcriptional silencing of reporter genes inserted next to, or within, centromeric repeats (Muller, 1930; Allshire *et al*, 1995; Festenstein *et al*, 1996).

The fission yeast *Schizosaccharomyces pombe* provides a paradigm for dissecting heterochromatin assembly because heterochromatin is not essential for cell viability and its minimal architecture closely resembles that of metazoa. Heterochromatin domains are associated with the *S. pombe* centromeres, telomeres and the mating-type locus, and are necessary for the functional integrity of these loci (Grewal, 2010). At centromeres, outer repeat sequences, composed of *dg* and *dh* elements, are assembled in heterochromatin. Fragments of the *dg* element (e.g. L5) are sufficient to form heterochromatin domains when placed at an ectopic locus (Partridge *et al*, 2002; Sadaie *et al*, 2004; Wheeler *et al*, 2009). The DNA sequence of all centromeric *dg* and *dh* elements is almost identical; however, the number and organization of these repeats vary between the three centromeres (Allshire, 2003). The similarity of centromere repeat sequences precludes the identification of minimal modules critical for heterochromatin assembly. These arrays of heterochromatin surround the central domain where CENP-A[Cnp1] replaces the histone H3 and the kinetochore forms. Heterochromatin and CENP-A[Cnp1] chromatin are both required to form functional centromeres (Buscaino *et al*, 2010; Grewal, 2010).

Fission yeast episomal plasmids require only part of an outer repeat plus an entire central domain DNA to form functional centromeres on minichromosomes. Such minichromosomes provide a powerful tool to dissect the

contribution of *dg* and *dh* elements to heterochromatin and kinetochore assembly (Baum *et al*, 1994; Folco *et al*, 2008). Moreover, the status of H3K9 methylation on these mini-chromosomes can be specifically monitored because the plasmid-borne outer repeat fragment is flanked by unique plasmid sequences. Mutations of a variety of factors disrupt heterochromatin integrity on these minichromosomes resulting in their instability and loss (Folco *et al*, 2008).

Convergent transcription within the *dg* and *dh* elements by RNA polymerase II (RNAPII) generates double-stranded RNA (dsRNA) that elicits an RNA interference (RNAi) response (Volpe *et al*, 2002; Djupedal *et al*, 2005; Kato *et al*, 2005). Dicer (Dcr1) ribonuclease cleaves these dsRNAs into short-interfering RNAs (siRNAs) that guide the Argonaute (Ago1)-containing RITS complex to homologous nascent transcripts by sequence complementarity (Verdel *et al*, 2004). Chromatin-associated RITS recruits the complex containing the histone H3K9 methyltranferase Clr4 (equivalent to metazoan Suv39/KMT1) to centromeric repeats (Zhang *et al*, 2008). Methylation of H3K9 by Clr4 provides binding sites for the chromodomain proteins Swi6$^{HP1}$, Chp1, Chp2 and Clr4 itself resulting in the formation of heterochromatin (Bannister *et al*, 2001; Sadaie *et al*, 2004; Petrie *et al*, 2005; Zhang *et al*, 2008).

The hypoacetylated state of histones that typifies hetero-chromatin involves three histone deacetylases (HDACs): Clr3, Clr6 and Sir2 (Grewal *et al*, 1998; Nakayama *et al*, 2001; Shankaranarayana *et al*, 2003; Freeman-Cook *et al*, 2005; Wiren *et al*, 2005; Yamada *et al*, 2005; Nicolas *et al*, 2007; Sugiyama *et al*, 2007). Clr3 is a component of the SHREC complex that physically interacts with the chromodomain protein Chp2 and Swi6$^{HP1}$. Clr3 deacetylates histone H3 on lysine 14 and limits access of RNAPII to centromeres (Sugiyama *et al*, 2007; Sadaie *et al*, 2008; Fischer *et al*, 2009). The Clr6 HDAC is incorporated into two distinct complexes that deacetylate several lysines on histone H3 and H4, particularly at the promoters and over the coding regions of genes (Wiren *et al*, 2005; Nicolas *et al*, 2007). Sir2 belongs to the Sirtuin family of HDACs that utilize NAD$^+$ as a cofactor (Rusche *et al*, 2003). *In vivo S. pombe* Sir2 preferentially deacetylate histone H3K9 (Shankaranarayana *et al*, 2003; Wiren *et al*, 2005). Cells lacking Sir2 display only partial defects in centromeric heterochromatin integrity and retain Swi6$^{HP1}$ localization at centromeres (Shankaranarayana *et al*, 2003; Freeman-Cook *et al*, 2005).

At fission yeast centromeres, telomeres and mating-type region, RNAi is required to establish heterochromatin (Hall *et al*, 2002; Sadaie *et al*, 2004; Verdel *et al*, 2004). Although H3K9me completely covers the centromeric outer repeats, RNAi and the resulting siRNAs are confined to specific regions within these repeats (Cam *et al*, 2005; Buhler *et al*, 2008; Djupedal *et al*, 2009; Halic and Moazed, 2010; Zaratiegui *et al*, 2011). It is unknown how H3K9me is established over regions of the outer centromeric repeats that are not targeted by RNAi. Moreover, at all three heterochromatin loci, RNAi is partly or completely dispensable for maintenance of H3K9me (Jia *et al*, 2004; Sadaie *et al*, 2004; Kanoh *et al*, 2005; Hansen *et al*, 2006; Partridge *et al*, 2007; Halic and Moazed, 2010).

In this study, we set out to uncover the mechanisms governing the assembly of large chromatin domains. We demonstrate that heterochromatin is first established at the siRNA-rich regions over nucleation sites containing RNAPII activity. Moreover, *de novo* heterochromatin estab-lishment assays unearth a role for the HDACs Sir2 Clr3 and the chromodomain protein Swi6$^{HP1}$ in, first, initiating the formation of heterochromatin and, then, in the mechanism that 'spreads' H3K9me from nucleation sites over neighbour-ing chromatin. Our analyses reveal that once heterochroma-tin has been established, its propagation is dependent on the parallel actions of the HDACs Sir2 and Clr3. This newly identified role for Sir2 in maintaining H3K9me-dependent heterochromatin is underscored by our finding that tethering Sir2 next to a heterochromatin nucleation site ensures hetero-chromatin maintenance in cells lacking RNAi. The HDAC-dependent pathway uncovered here aids the RNAi pathway to propagate fully assembled and functional heterochromatin domains. The analyses presented provide the first clear evidence for a sequential assembly mechanism required to form intact heterochromatin domains at centromeres.

## Results

### *Defining heterochromatin nucleation sites within the centromeric dg elements*

Inducers of specific epigenetic states are frequently only required for the initiation but not for the preservation of that state (Berger *et al*, 2009). *De novo* establishment assays have previously demonstrated that RNAi is required to nucleate heterochromatin in fission yeast (Hall *et al*, 2002; Jia *et al*, 2004; Sadaie *et al*, 2004). At centromeres, genome-wide analyses have shown that H3K9me covers the entire outer repeat region, which is composed of *dg–dh* elements (Cam *et al*, 2005; Zaratiegui *et al*, 2011). In contrast, siRNA profiling analyses demonstrate that the vast majority of siRNAs are derived from restricted regions within the *dg* and *dh* elements. We refer to these siRNA hotspots as 'siRNA-rich', as opposed to the remaining 'siRNA-void' region (Figure 1A and Supplementary Figure S1; (Cam *et al*, 2005; Buhler *et al*, 2008; Djupedal *et al*, 2009; Halic and Moazed, 2010; Zaratiegui *et al*, 2011). The restriction of siRNAs to specific regions suggests that this confined RNAi activity may create heterochromatin nucleation centres from which heterochromatin expands. Alternatively, the siRNA-void regions may nucleate heterochromatin independently of RNAi. However, understanding the contribution of a particular centromeric DNA element to heterochromatin formation is problematic because of the repetitive nature of outer repeats (Supplementary Figure S2A). Indeed, to date, all genome-wide chromatin immunoprecipitation (ChIP) ana-lyses of heterochromatin components in *S. pombe* only indicates the average distribution since the signal intensity is normalized to the number of repeats (Cam *et al*, 2005; Zaratiegui *et al*, 2011). This normalization is performed because the signal cannot be assigned to any specific *dg*/dh element. Furthermore, even PCR primers, designed using the current genome database, that are predicted to allow the amplification of products unique to the *dg*/*dh* arrangement at centromere 1 (*cen1*) can amplify a PCR product from cells completely lacking *cen1* (Supplementary Figures S2A and B;(Ishii *et al*, 2008). This suggests that the available centromeric repeat sequence and organization is inaccurate and requires further exploration. These considerations

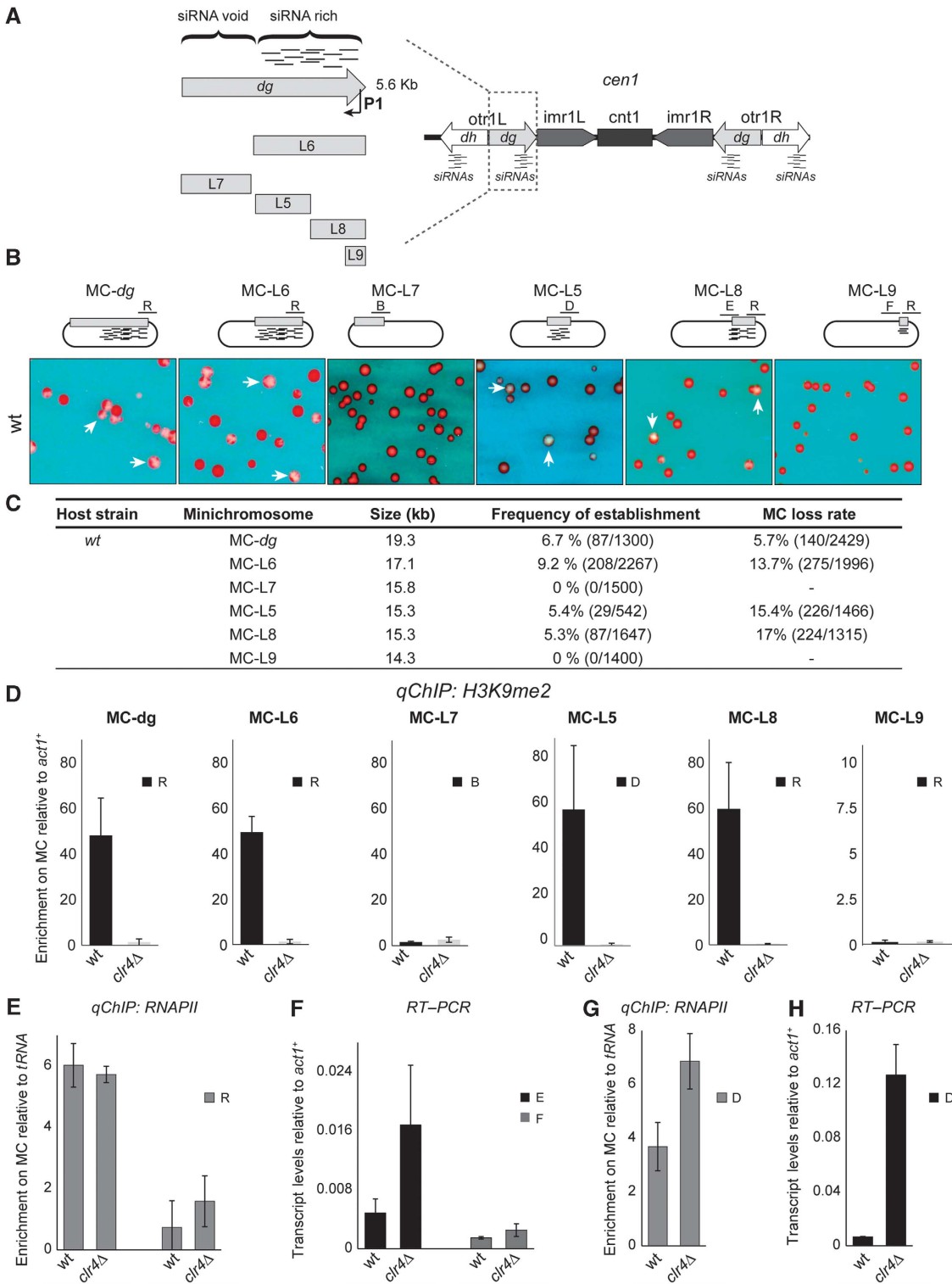

Figure 1 Heterochromatin establishment over the centromeric *dg* fragment. (**A**) Schematic of fission yeast *cen1* (right panel), the described P1 promoter (Djupedal *et al*, 2005) and nomenclature of the different *dg* fragments analysed in this study (left panel). (**B**) Colony colour assay to assess minichromosome stability. Wt cells transformed with MC-*dg*, L6, L7, L5, L8 and L9 were plated on limiting adenine plates. Red colonies indicate unstable minichromosomes; white/sectored colonies (white arrows) are indicative of proper segregation at mitosis. (**C**) Establishment frequency and loss rate of indicated plasmid-based minichromosomes in wt strain. (**D**) Quantitative ChIP (qChIP) to detect H3K9me2 levels associated with MC-*dg*, L6, L7, L5, L8 and L9 fragments upon transformation into wt or *clr4Δ* cells. Specific primers were used to analyse the enrichment on minichromosome (MC) relative to actin (*act1*$^+$). (**E**) qChIP to detect RNAPII on L8- and L9-containing plasmids (MC-L8 and MC-L9) relative to a tRNA gene in wt and *clr4Δ* cells. (**F**) qRT–PCR to detect transcripts originating from the L8 and L9 fragments relative to *act1*$^+$ in wt and *clr4Δ* cells. (**G**) Quantitative chromatin immunoprecipitation to detect RNAPII on L5-containing plasmid (MC-L5) relative to a tRNA gene in wt and *clr4Δ* cells. (**H**) qRT–PCR to detect transcripts originating from the L5 fragment relative to *act1*$^+$ in wt and *clr4Δ* cells. Error bars in (**D**) to (**H**): s.d. of three biological replicates.

prompted us to use minichromosomes to characterize specific centromere sequence elements.

Episomal plasmids bearing outer repeat regions plus a central domain have been shown to assemble functional centromeres that result in mitotically stable minichromosomes (Baum *et al*, 1994). These plasmids must establish and maintain heterochromatin, otherwise centromere function and minichromosome stability is compromised. A colony colour-sectoring assay is used to monitor plasmid retention (white) and loss (red). Using this plasmid-based heterochromatin establishment assay, we directly tested whether the siRNA-rich region can nucleate heterochromatin. Minichromosome plasmids (MC) bearing a full-length *dg* or *dg* fragments were transformed into wild-type (wt) or *clr4* null cells (*clr4*Δ, completely devoid of heterochromatin) (Figure 1A, Supplementary Figure S2C). As expected, plasmids bearing full-length *dg* (MC-*dg*) form mitotically stable minichromosomes in wt cells (white colonies with red sectors) at a frequency of 6.7% but not in *clr4*Δ cells (red colonies) (Figures 1B and C and Supplementary Figure S2D). Importantly, the MC-L6 minichromosome, containing the siRNA-rich *dg* subfragment, established functional centromeres in *wt* (9.2%) but not in *clr4*Δ cells. In contrast, MC-L7, which contains only the siRNA-void region, was unable to establish functional centromeres (0%; Figures 1B and C and Supplementary Figure S2D).

To further test if plasmid stability correlates with the establishment of heterochromatin on minichromosomes, the presence of H3K9 methylation was assessed by ChIP. Since the minichromosome-borne *dg* fragments are in a unique sequence context relative to endogenous repeats, the *dg*–plasmid junction can be specifically monitored. In *wt*, but not *clr4*Δ cells, high levels of H3K9me2 were detected on MC-*dg* and MC-L6 (Figure 1D). No H3K9 methylation was detected on the siRNA-void L7 fragment (MC-L7; Figure 1D). Thus, the siRNA-rich *dg* L6 region acts as a nucleation site that seeds heterochromatin formation. In contrast, the L7 siRNA-void region is unable to establish H3K9me: heterochromatin must spread into these sequences from flanking nucleation sites.

### RNAPII activity defines multiple redundant nucleation sites within the siRNA-rich region

An RNAPII promoter resides within the L6 fragment of the *dg* element (Djupedal *et al*, 2005; P1, Figure 1A and Supplementary Figure S3). To test the role of this promoter in heterochromatin establishment, minichromosomes containing L6 subfragments were tested for their ability to nucleate heterochromatin (Figure 1A, Supplementary Figure S2C). The L8 fragment contains the active P1 promoter as confirmed by RNAPII ChIP and RT–PCR analyses (Figures 1E and F). The L8 fragment confers mitotic stability in *wt* (5.3%), but not *clr4*Δ, cells (MC-L8; Figures 1B and C, Supplementary Figures S2D and S3). In contrast, the L9 subfragment, lacking the P1 promoter TATA box and lacking RNAPII (Figures 1E and F), is unable to form functional centromeres on minichromosomes and is thus mitotically unstable (0% MC-L9; Figures 1B and C, Supplementary Figures S2D and S3).

ChIP analyses confirm that H3K9me can be established by the L8, but not the L9, subfragment (MC-L8 and MC-L9; Figure 1D). The L5 subfragment also lacks the P1 promoter (Figure 1A); however, it attracts high levels of H3K9me2 and

imparts mitotic stability to MC-L5 (Figures 1B–D). ChIP for RNAPII demonstrated that it is enriched on the L5 fragment in both *wt* and *clr4*Δ cells (Figure 1G). Furthermore, RT–PCR and 5′ RACE analyses allowed the detection of transcripts that originate from both strands within the L5 element, thus revealing the presence of additional RNAPII promoters (Figure 1G and Supplementary Figure S4A).

The above analyses indicate that the siRNA-rich region (L6) is modular and contains at least two independent regions capable of nucleating heterochromatin. Each of these nucleation sites contains RNAPII promoter activity, suggesting that localized RNAPII transcription and RNAi activity seed heterochromatin assembly within *dg*, whereas the siRNA-void region alone is unable to nucleate heterochromatin.

### HDACs and Swi6[HP1] are required for the de novo spreading of a heterochromatin domain

The formation of large heterochromatin domains over an entire outer repeat suggests a mechanism that promotes its spreading from siRNA-rich nucleation sites into siRNA-void regions. To identify components important for establishment and/or spreading of H3K9 methylation, MC-*dg* was transformed into *wt* cells and several mutants known to partially disrupt heterochromatin (Figure 2A). H3K9me2 ChIP was performed to assess heterochromatin formation close to the siRNA-rich nucleation region (PCR R) or near the siRNA-void region (PCR V) (Figure 2A). High levels of H3K9me were detected on both regions of the *dg* element in *wt* and *pst2*Δ cells (Pst2 is a specific component of HDAC Clr6 complex II; (Nicolas *et al*, 2007)), suggesting that Clr6 complex II makes only a limited contribution to heterochromatin nucleation or spreading (Figure 2B). However, no H3K9 methylation was detected on either the V or R regions in cells lacking Dicer (*dcr1*Δ) confirming that RNAi is absolutely required for targeting heterochromatin to centromere repeats (Figure 2B). When MC-*dg* was transformed into cells lacking either Sir2 or Swi6[HP1], a strikingly asymmetric distribution of H3K9me was evident. H3K9me2 was clearly established over the nucleation site (R), but not at the siRNA-void region (V). A similar pattern was also observed in *clr3*Δ cells, except that a low level of H3K9me was detected at the siRNA-void region (Figure 2B).

Thus, the HDACs, Clr3 and Sir2, along with the architectural component Swi6[HP1], are required to spread H3K9 methylation from siRNA-rich nucleation sites into flanking regions. This supports a model in which the *de novo* assembly of heterochromatin at centromeres involves a two-step mechanism: establishment at nucleation sites specified by RNAi and subsequent spreading mediated by Sir2, Clr3 and Swi6[HP1].

### Two distinct types of nucleation sites reside within the siRNA-rich region of centromere repeats

To further characterize HDACs function in heterochromatin nucleation and spreading, we analysed the H3K9me pattern obtained upon introducing different minichromosomes into *sir2*Δ cells. When MC-*dg* was transformed in *wt* cells, H3K9me was enriched on both V and R regions but not detected at V in *sir2*Δ cells, or at V or R regions in *clr4*Δ cells (Figures 3A and B). The L6 subfragment allows H3K9me on both sides in *wt* cells; however, in *sir2*Δ cells high levels of H3K9me were detected on the right (PCR R), but not the left

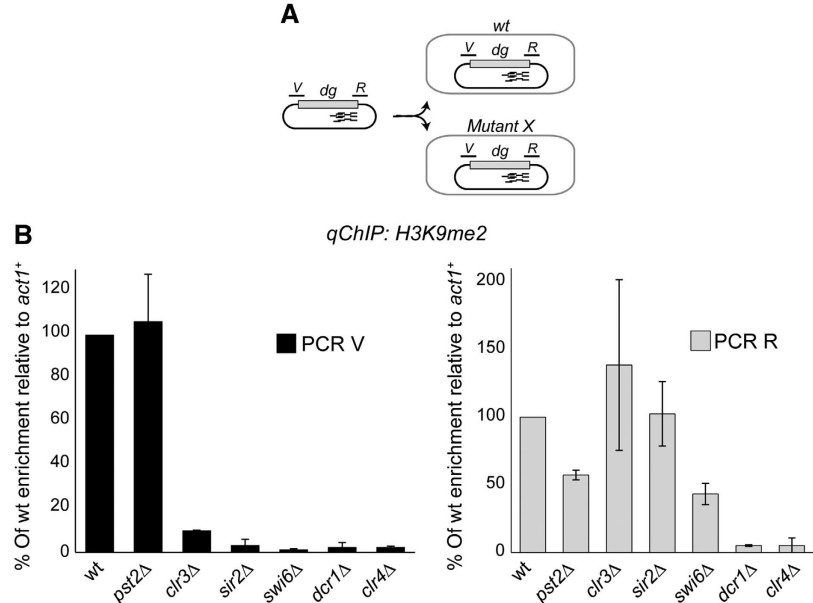

**Figure 2** Heterochromatin spreading over the centromeric *dg* fragment. (**A**) Diagram of procedure to assess heterochromatin establishment on MC-*dg* siRNA-rich (R) and siRNA-void (V) regions upon transformation into wt or mutant cells. (**B**) qChIP to assess H3K9me2 levels associated with the MC-*dg* siRNA-rich (R) and siRNA-void (V) regions. Enrichment is shown relative to actin (*act1*[+]), and normalized to wt. Error bars: s.d. of three biological replicates.

side (PCR A, MC-L6; Figures 3A and B). The L5 subfragment displayed a similar pattern as L6, H3K9me was only detected on the right side in the absence of Sir2 (PCR D, MC-L5; Figures 3A and B). In contrast, no H3K9me occurred on either side of the L8 fragment in *sir2*Δ cells, even though MC-L8 efficiently attracts H3K9me in *wt* cells (MC-L8; Figures 3A and B). Similar dependencies were observed when these DNA fragments were tested in *clr3*Δ and *swi6*Δ cells, with the exception that H3K9me could be established on both sides of the L5 nucleation region (MC-L5 and MC-L8; Figure 3C). The defects observed in heterochromatin nucleation in *sir2*Δ, *clr3*Δ and *swi6*Δ cells is unlikely due to defective RNAi since centromeric siRNAs are still produced in these mutants (Buhler *et al*, 2006; Sugiyama *et al*, 2007) and Supplementary Figure S4B). These analyses indicate that the siRNA-rich region encompasses two distinct types of nucleation site: the L5 region, which allows H3K9 methylation independently of Sir2 HDAC and Swi6[HP1], and the L8 region, which requires both Sir2 and Swi6[HP1] to form heterochromatin. These different types of nucleation sites must act together within centromere repeats providing redundant processes that ensure robust heterochromatin assembly.

### Sir2, Clr3 and Swi6[HP1] function in parallel to RNAi to maintain H3K9 methylation and heterochromatin function on centromere repeats

Heterochromatin domains are required to ensure full centromere function and accurate chromosome segregation (Ekwall *et al*, 1995; Bernard *et al*, 2001; Nonaka *et al*, 2002; Volpe *et al*, 2003). We next investigated whether factors required for heterochromatin nucleation (i.e., RNAi) and spreading (Sir2, Clr3 and Swi6[HP1]) contribute separately to the maintenance of centromeric heterochromatin.

The silencing of genes placed within centromeric heterochromatin provides a sensitive readout of heterochromatin

integrity (Allshire *et al*, 1995). In the absence of RNAi (*dcr1*Δ), H3K9me (*clr4*Δ) or associated architectural components, such as Swi6[HP1] (*swi6*Δ), silencing of an *ade6*[+] reporter gene within the *dg* repeat of *cen1* (*cen1-dg:ade6*[+]) is lost (white colonies; Ekwall *et al*, 1999). However, *cen1-dg:ade6*[+] silencing is largely unaltered in *sir2*Δ cells (red colonies), or in cells expressing a partially defective RNAi component (*cid12-ha*; subunit of the RDRC complex; Motamedi *et al*, 2004). When *cid12-ha*; was combined with *sir2*Δ or *clr3*Δ, a synergistic loss of *cen1-dg:ade6*[+] silencing was observed (pink/white colonies; Figure 4A). Moreover, when *dcr1*Δ was combined with *sir2*Δ, *clr3*Δ or *swi6*Δ increased sensitivity to the microtubule-destabilizing compound thiabendazole (TBZ) was observed (Figure 4B). Elevated TBZ sensitivity indicates that centromere function is more defective in these double mutants. To directly test the contribution of various components to heterochromatin-associated centromere function, we again exploited the minichromosome system. A minichromosome bearing full-length *dg* repeat (MC-*dg''*; Baum *et al*, 1994) was first transformed into *wt* cells to establish the full heterochromatin domain and subsequently it was crossed into specific mutants (Figure 4C). MC-*dg''* centromere function and H3K9 methylation is retained in wt and in *sir2*Δ but not in *clr4*Δ progeny (Folco *et al*, 2008) and Figures 4D–F). MC-*dg''* mitotic stability was reduced, but not obliterated, in *dcr1*Δ and *ago1*Δ progeny (Figures 4D and E). In agreement with this, H3K9me2 was detected on MC-*dg''*, in both *dcr1*Δ and *ago1*Δ cells, with higher levels over the siRNA-void region relative to the siRNA-rich region (PCR V and R; Figure 4F). In contrast, following its transmission into *dcr1*Δ*sir2*Δ double-mutant cells the MC-*dg''* completely lost centromere function (only red colonies; Figures 4D and E). Moreover, as in *clr4*Δ cells, no H3K9me2 was detectable on either the siRNA-void or -rich regions of the *dg* element (PCR V and R; Figure 4F).

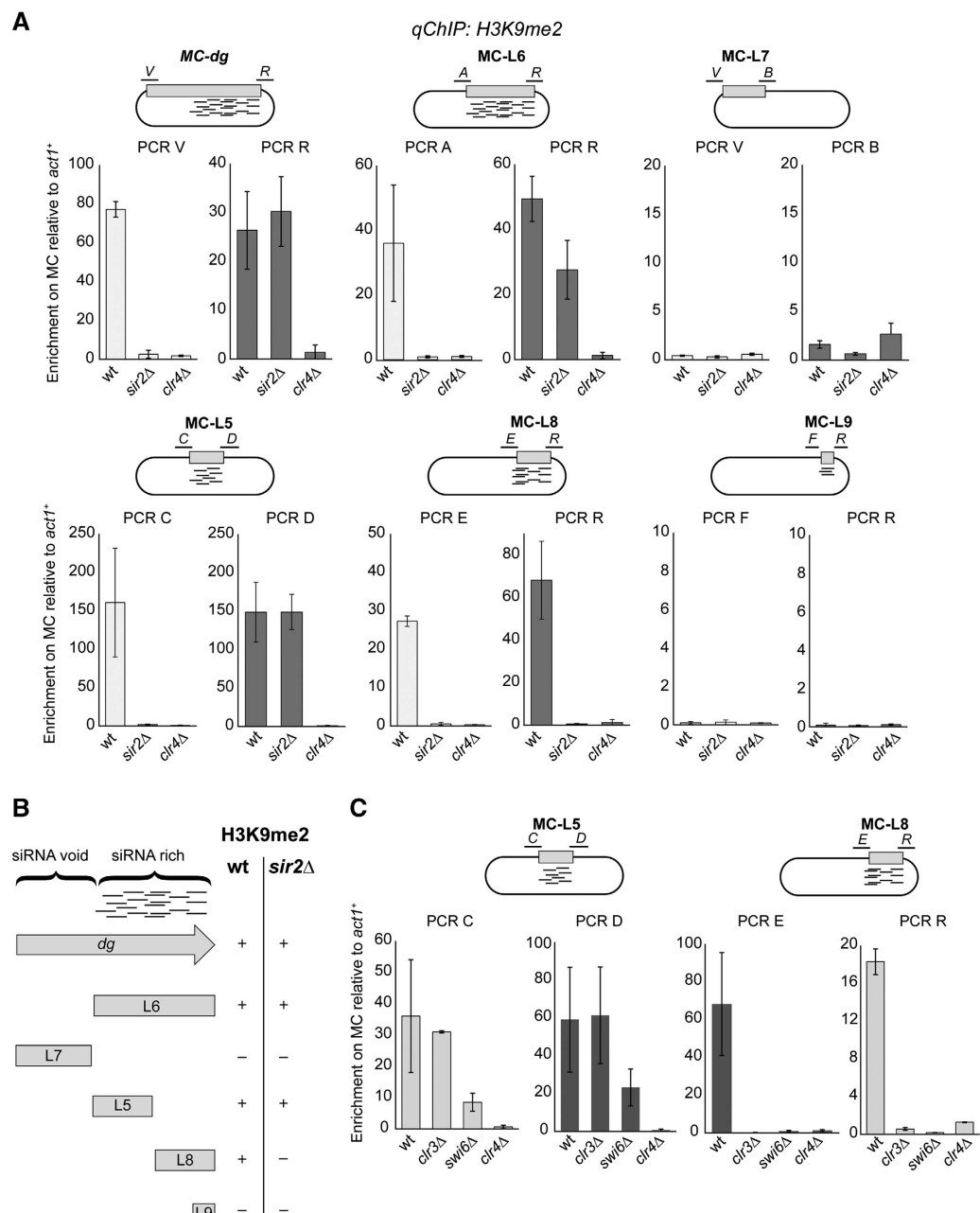

**Figure 3** Heterochromatin nucleation over the centromeric *dg* subfragments in wt, *sir2Δ*, *swi6Δ* and *clr3Δ* cells. (**A**) qChIP to assess H3K9me2 levels associated with the right and left sides of indicated MC upon transformation into wt, *sir2Δ* and *clr4Δ* cells. Enrichment is shown relative to actin (*act1*[+]), and normalized to wt. (**B**) Schematic diagram to summarize MC ability to nucleate heterochromatin in wt and *sir2Δ* cells. (**C**) qChIP to assess H3K9me2 levels associated with the right and left sides of MC-L5 and MC-L8 upon transformation into wt, *clr3Δ*, *swi6Δ* and *clr4Δ* cells. Enrichment is shown relative to actin (*act1*[+]), and normalized to wt. Error bars: s.d. of three biological replicates in (**A**) and (**C**).

These analyses demonstrated that the RNAi pathway collaborates with the HDACs Sir2 and Clr3, and with Swi6[HP1], to maintain heterochromatin-associated centromere functions on *dg* repeats.

We next assessed if RNAi also operates synergistically with Sir2, Clr3 and Swi6[HP1] on endogenous centromere to maintain H3K9 methylation. As shown in Figure 4G, H3K9me2 levels are detectable but reduced in *dcr1Δ*, *ago1Δ* and *sir2Δ* relative to *wt* cells. However, H3K9me2 was completely lost from centromeric repeats in both *dcr1Δsir2Δ* and *dcr1Δswi6Δ* double mutants while still detected in *dcr1Δago1Δ* double-mutant cells (Figure 4G and Supplementary Figure S4C).

Similarly, H3K9 methylation has been shown to be strongly reduced at centromeres in *dcr1Δclr3Δ* double mutants (Yamada *et al*, 2005). H3K9me creates binding sites for the chromodomain protein Swi6[HP1]. Immunolocalization analyses reveal that GFP-tagged Swi6 remains associated with centromeres in *dcr1Δ* and *sir2Δ* cells (Ekwall *et al*, 1999; Freeman-Cook *et al*, 2005). However, similar to *clr4Δ* cells, in *dcr1Δsir2Δ* double mutants the localization of GFP-Swi6 at centromeres is completely lost (Figure 4H). This was confirmed by ChIP analyses (Figure 4I).

We conclude that, at centromeres, after the initial establishment of a heterochromatin domain, RNAi becomes dispen-

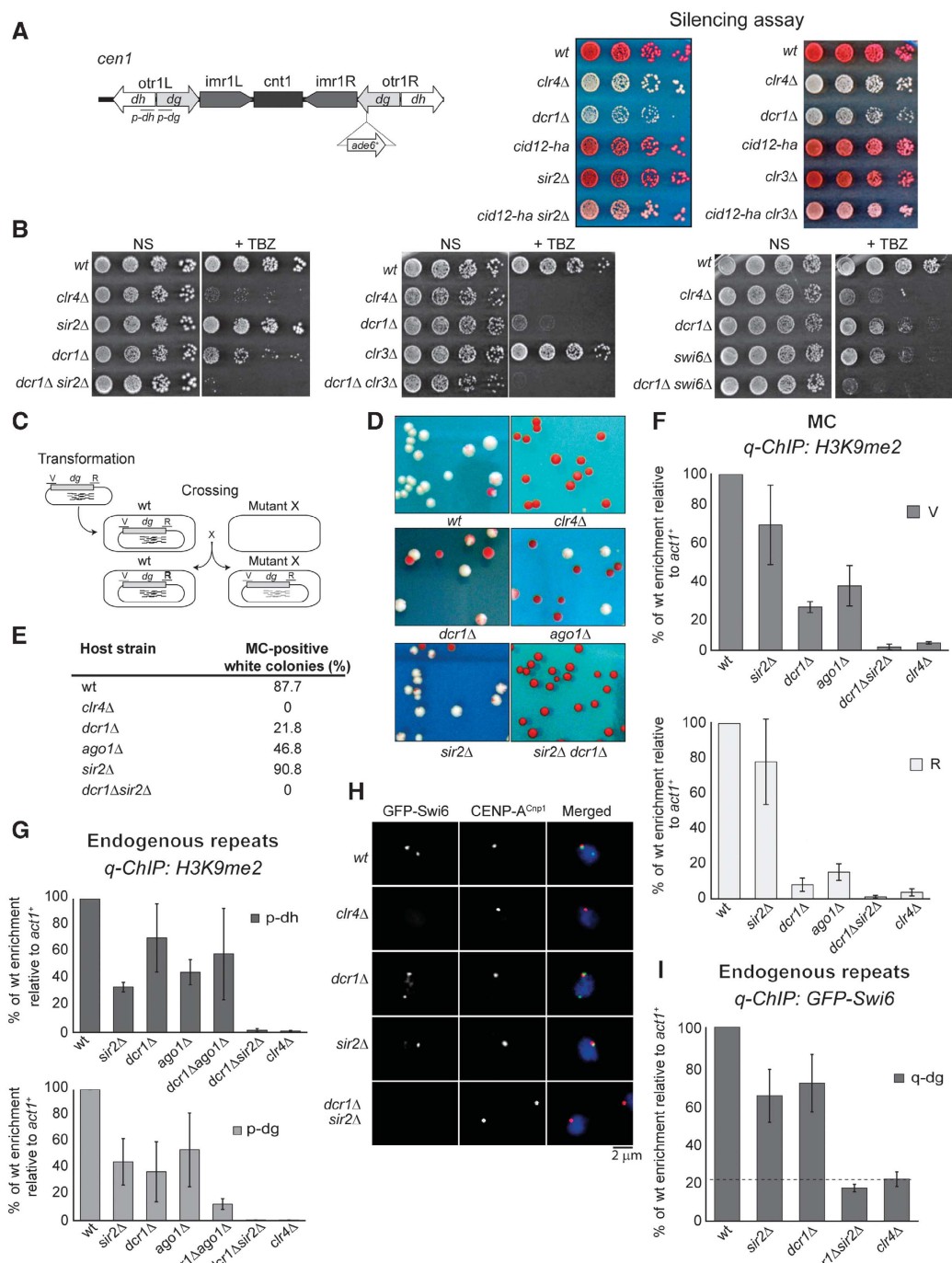

**Figure 4** Sir2, Clr3 and Swi6[HP-1] are required for heterochromatin maintenance in RNAi-compromised cells. (**A**) Silencing assay at centromere *otr1R(SphI):ade6+*. Left panel: diagram indicating *ade6+* reporter gene insertion at *dg* of cen1. The position of PCR products (*p-dg* and *p-dh*) is also indicated. The primers also hybridize with centromere 2 and 3 (Supplementary Figure S2A). Right panel: serial dilutions of cells were spotted onto limiting adenine medium. In wt cells, *ade6+* is repressed (red colonies); silencing-compromised mutants alleviate the repression (pink/white colonies). (**B**) Serial dilutions of cells were spotted onto non-selective (NS) plates or 10 μg/ml TBZ-containing medium. (**C**) Diagram of procedure to assess the heterochromatin maintenance on minichromosome containing two *dg* elements (MC-*dg″*) in wt and mutant cells. R and V indicate regions on minichromosome analysed by qPCR. (**D**) Colony colour assay to assess minichromosome stability. Cells containing MC-*dg″* were crossed into wt and indicated mutants, and plated on limiting adenine plates. Red colonies indicate unstable minichromosomes; white/sectored colonies indicate stable minichromosomes that are retained at mitosis. (**E**) Percentage of MC-positive white/sectored colonies in indicated host strains. (**F**) qChIP analyses of H3K9me2 levels maintained on MC-*dg″* following crosses into wt and indicated mutant strains. Enrichment is shown relative to actin (*act1+*), and normalized to wt. R and V correspond to regions on minichromosome analysed by qPCR. (**G**) qChIP analyses of H3K9me2 levels associated with endogenous centromere *dh* (top; *p-dh*) and *dg* (bottom; *p-dg*) elements. Enrichment is shown relative to actin (*act1+*), and normalized to wt. (**H**) Immunofluorescence analysis of GFP-Swi6 in wt or mutant cells. Representative images show staining of GFP-Swi6 (green), CENP-A[Cnp1] (red) and DNA (DAPI-blue). (**I**) qChIP analyses of GFP-Swi6 levels associated with endogenous centromere *dg* repeats. Enrichment is shown relative to actin (*act1+*), and normalized to wt. Error bars for (**F**), (**G**) and (**I**): s.d. of three biological replicates.

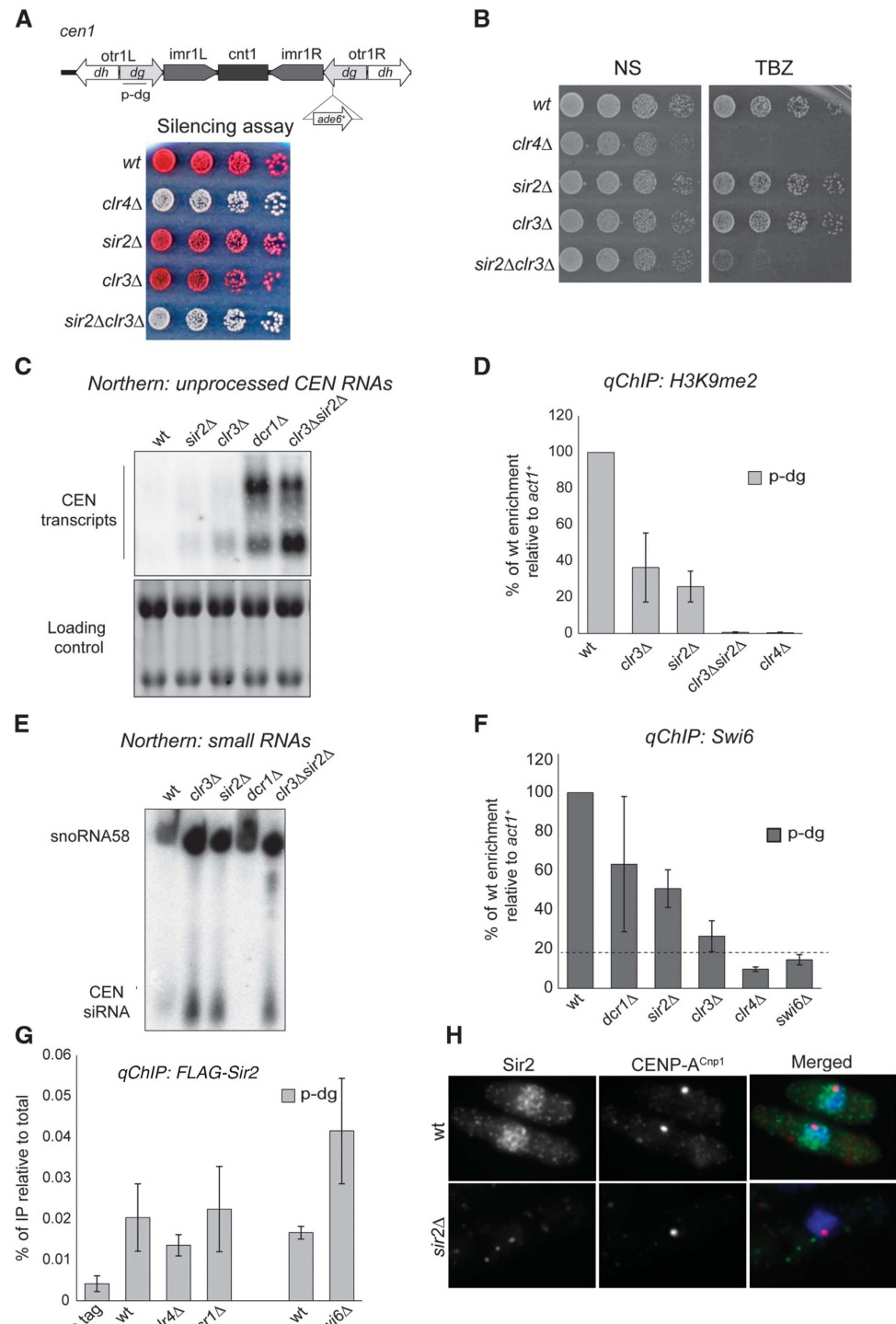

**Figure 5** Sir2 and Clr3 are essential for propagating H3K9me2. (**A**) Top: diagram indicating the position of PCR products (p-dg) on cen1. The primers also hybridize with centromere 2 and 3 (Supplementary Figure S2A). Bottom: serial dilutions of cells were spotted onto limiting adenine medium. In wt cells, *ade6*[+] is repressed (red colonies); silencing-compromised mutants alleviate the repression (pink/white colonies). (**B**) Serial dilutions of cells were spotted onto non-selective (N) plates or 10 g/ml TBZ-containing medium. (**C**) Northern: unprocessed *otr* transcripts in wt, and indicated mutants. Loading control: rRNA. (**D**) qChIP analyses of H3K9me2 levels associated with endogenous centromere *dg* (p-dg) in wt and indicated mutant background. Enrichment is shown relative to actin (*act1*[+]), and normalized to wt levels. (**E**) Northern: centromeric siRNAs in wt, and indicated mutants. Loading control: snoRNA58. (**F**) qChIP analyses of Swi6[HP1] associated with endogenous centromere *dg* (*p-dg*) in wt and indicated mutants background. Enrichment is shown relative to actin (*act1*[+]), and normalized to wt. (**G**) qChIP analyses of FLAG-Sir2 associated with endogenous centromere *dg* (p-dg) in wt and indicated mutants background. Error bars: s.d. of three biological replicates in (**D**), (**F**) and (**G**). (**H**) Immuno-localization analysis of Sir2 and CENP-A[Cnp1] in wt or *sir2*Δ cells. Representative images show staining of Sir2 (green), CENP-A[Cnp1] (red) and DNA (DAPI-blue).

sable for its retention and that Sir2, Clr3 and Swi6[HP1] can independently propagate the remaining heterochromatin.

### Maintenance of H3K9 methylation at centromeres requires the HDACs Sir2 and Clr3, even in the presence of active RNAi

Sir2, Clr3 and Swi6[HP1] are clearly required along with the RNAi pathway to maintain centromeric heterochromatin. However, the two HDACs may act together in the same pathway or separately.

To distinguish between these two possibilities, we analysed centromeric heterochromatin integrity in *sir2Δclr3Δ* double-mutant cells. As shown in Figure 5A, silencing of *cen1-dg:ade6+* marker gene was alleviated in *sir2Δclr3Δ* double-mutant cells but not in the correspondent single mutants. In contrast to *sir2Δ* and *clr3Δ* single mutants, *sir2Δclr3Δ* double-mutant cells display high TBZ sensitivity (Figure 5B) suggesting defects in heterochromatin integrity at centromeres. Indeed, in contrast to the single mutants, long unprocessed centromeric transcripts accumulate in *sir2Δclr3Δ* double-mutant cells and no H3K9me2 is maintained at centromeres (Figures 5C and D). This demonstrates that Sir2 and Clr3 promote heterochromatin integrity independently. High levels of *cen*-siRNAs are produced in the absence of Sir2 and Clr3 demonstrating that the presence of one of these two HDACs is essential for propagating H3K9me2 eventhough RNAi remains active (Figure 5E). The chromodomain protein Swi6[HP1] physically interacts with Clr3 suggesting that these two proteins cooperate to maintain heterochromatin (Yamada *et al*, 2005). In agreement with this finding, we find that Swi6[HP1] association with centromeric repeats is severely compromised in *clr3Δ* cells (∼10% of wt relative to background in *swi6Δ*) while a reduction to only 50 and 40% occurs in the absence of Dcr1 or Sir2, respectively (Figure 5F).

Sir2 behaviour is distinct. We find that the association of Sir2 with centromeric repeats does not require Swi6[HP1], the RNAi component Dcr1, or the Clr4 methyltransferase (Figure 5G). Therefore, Sir2 interacts with centromeric repeats independently of the other activities that coalesce to assemble heterochromatin. In addition, Sir2 is also detected at many other genomic locations that do not assemble heterochromatin (Wiren *et al*, 2005; Supplementary Figure S4E), and Sir2 has a diffuse nuclear localization (Figure 5H). These findings demonstrate that Sir2 and Clr3 contribute to H3K9 methylation maintenance via distinct pathways. We propose that Sir2 and Clr3 independently suppress transcription originating from centromere repeats. In the absence of both these HDACs, high transcriptional activity prevents H3K9 methylation propagation, even in the presence of active RNAi.

### Sir2 is sufficient to maintain heterochromatin in the absence of RNAi

Our analyses demonstrate that a HDAC-dependent pathway acts to maintain heterochromatin in the absence of RNAi. The L5 fragment of *dg* mediates heterochromatin assembly when inserted at an ectopic locus (Partridge *et al*, 2002; Sadaie *et al*, 2004; Wheeler *et al*, 2009, 2012). L5-driven heterochromatin integrity is partially independent of the HDAC Sir2 (Supplementary Figure S5A).

In contrast, artificial tethering of TetR[off]-Sir2 at the *ura4* locus (ura4:*4xTetO-ade6+*) is not sufficient to induce

heterochromatin assembly confirming that RNAi activity is essential for establishing heterochromatin *de novo* (Supplementary Figure S5B). To test if the Sir2 HDAC activity is sufficient to maintain H3K9me in the absence of RNAi, we artificially tethered Sir2 adjacent to *dg*-L5 at the *ura4* locus where *dg*-L5-*4xTetO-ade6+* was inserted (Figure 6A). The *dg*-L5-*4xTetO-ade6+* allowed heterochromatin formation, as indicated by silencing of the *ade6+* (43.7% red/pink repressed colonies) and high levels of H3K9me2 (Figures 6B–E). However, in cells lacking RNAi (*dcr1Δ*), silencing of *ade6+* was alleviated (100% white colonies) and H3K9me2 was lost (Figures 6B–E). Thus, unlike endogenous centromere repeats, silencing of, and the formation of heterochromatin at, *dg*-L5-*4xTetO-ade6+* is completely dependent on RNAi.

Remarkably, in *dcr1Δ* cells, the presence of functional TetR[off]-Sir2 allowed silencing (45.6% red colonies) of *L5-4xTetO-ade6+* to persist, whereas it was lost when catalytically inactive TetR[off]–Sir2[N247A] was expressed (Figures 6B and C). Both the TetR[off]–Sir2 and TetR[off]–Sir2[N247A] fusion proteins were recruited to the TetO sites (Figure 6D). Furthermore, tethering of TetR[off]–Sir2, but not TetR[off]–Sir2[N247A], allowed the retention of high H3K9me2 levels in *dcr1Δ* cells (Figure 6E). The fact that the artificial tethering of Sir2 allows high levels of H3K9me2 in the absence of RNAi strongly supports the conclusion that Sir2 is necessary and sufficient for maintaining heterochromatin at centromeres.

We propose that Sir2, Clr3 and Swi6[HP1] are components of an epigenetic memory module that maintains heterochromatin independently of RNAi. RNAi initially acts to target Clr4 activity to centromere repeats. The resulting H3K9 methylation directly recruits the architectural component Swi6[HP1] thereby nucleating heterochromatin. Chromatin-associated Sir2 HDAC subsequently cooperates with Swi6[HP1], and the associated Clr3 HDAC, to induce and extend a hypoacetylated chromatin state that is essential for heterochromatin maintenance through subsequent cell division.

## Discussion

The assembly of large chromatin domains is generally thought to require: (i) initiation events, that establish the altered state at specific genomic locations; (ii) spreading, that involves the outwards expansion of this distinct chromatin state to coat adjacent chromosomal regions; and (iii) maintenance, to lock in the established state so that it is propagated long after the initiator has disappeared (Bonasio *et al*, 2010). Frequently, specific nucleation sites serve as entry points for chromatin modifiers allowing the initial formation of a specialized chromatin pocket and provide a seed for the subsequent spreading of the modified state *in cis*, independently of the underlying DNA sequence (Talbert and Henikoff, 2006). Here, we have dissected mechanisms governing centromeric heterochromatin formation and provided direct evidence to support the three-step model for the establishment and propagation of a distinct chromatin state (Figure 7).

### Nucleation sites direct the location of centromeric heterochromatin

We demonstrate that, at centromeres, H3K9 methylation is first initiated at nucleation sites corresponding to siRNA-rich segments. siRNAs are generated from non-coding centromeric

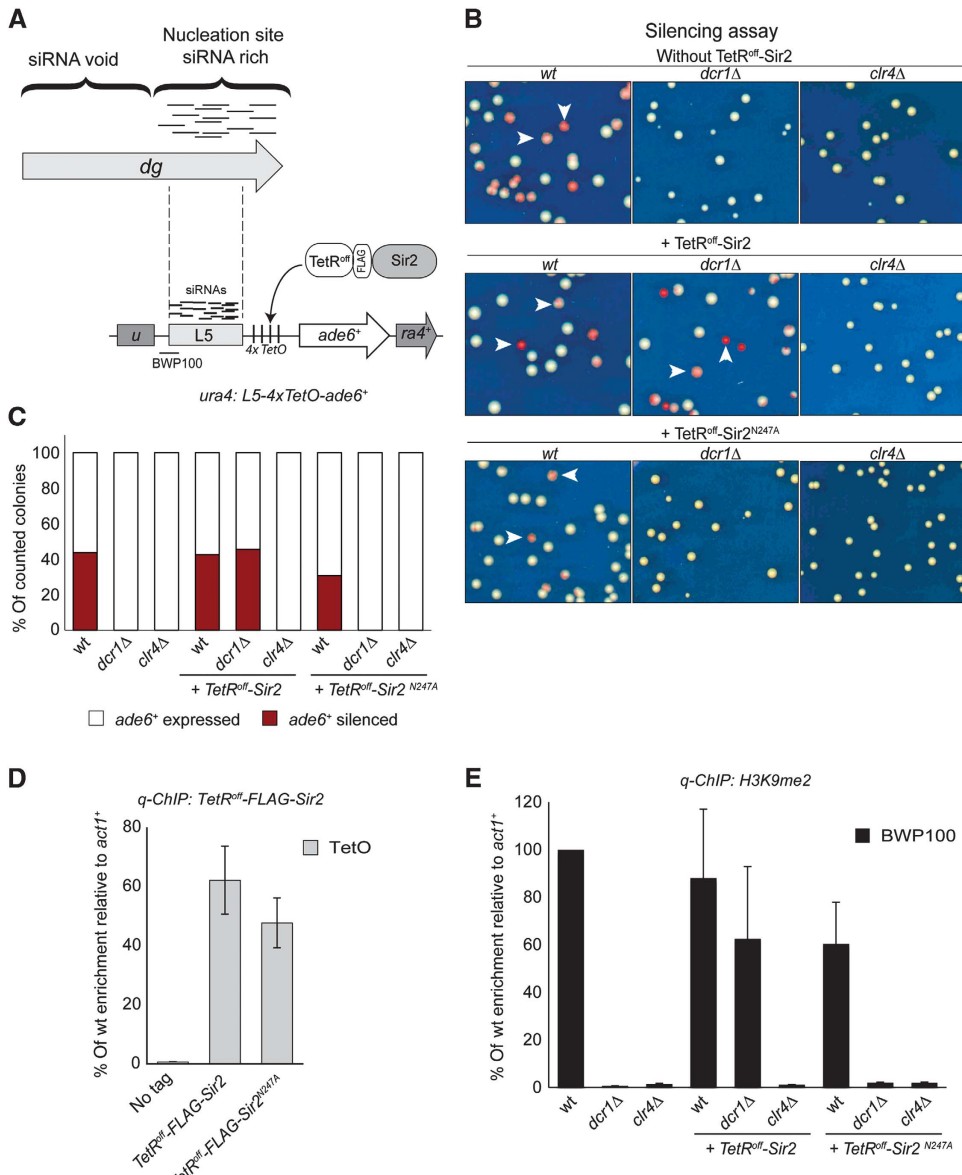

**Figure 6** Sir2 is sufficient to maintain heterochromatin in RNAi mutants. (**A**) Diagram of constructs used: the *L5-4xTetO-ade6*[+] reporter is inserted at the *ura4*[+] locus. TetR[off]-2 × FLAG-Sir2 is integrated at *leu1*[+] locus. The position of PCR products (BWP100 and TetO) on the *L5-4TetO-ade6*[+] reporter is indicated. (**B**) Silencing assay of *L5-4 × TetO-ade6*[+] in wt, *dcr1*Δ and *clr4*Δ cells without any tethered protein (top panel) or containing TetR[off]-2 × FLAG-Sir2 (middle panel) and TetR[off]-2 × FLAG-Sir2[N247A] (bottom panel). Cells were plated on medium with limiting adenine. Red/sectored colonies (arrows) indicate silencing of the *ade6*[+] reporter. (**C**) Quantification of *L5-4 × TetO-ade6*[+] silencing assay. (**D**) qPCR analyses of TetR[off]-2 × FLAG-Sir2 and TetR[off]-2 × FLAG-Sir2[N247A] levels associated with the *L5-4TetO-ade6*[+] reporter. Enrichment is shown relative to actin (*act1*[+]). (**E**) qChIP analyses of H3K9me2 levels associated with *L5-4xTetO-ade6*[+] reporter. Enrichment is shown relative to actin (*act1*[+]), and normalized to wt. Error bars for (**D**) and (**E**): s.d. of three biological replicates.

repeat transcripts and act as the inducers that home in on these sites, presumably by engaging homologous nascent transcripts. This then triggers the initial H3K9 methylation events by recruiting the Clr4 methyltransferase.

In other systems, sites of RNA production have also been shown to act as nucleation sites and non-coding RNAs are known to be required for the initial recruitment of chromatin modifiers to specific chromosomal regions (Herr and Baulcombe, 2004; Wutz, 2011; Conrad *et al*, 2012). In plants, production of siRNA can initiate RNA-directed DNA methylation and transcriptional silencing (Herr and Baulcombe, 2004). Dosage compensation mechanisms in flies and mammals also utilize non-coding RNAs to nucleate the recruitment of

chromatin modifiers that equalize the expression of X-linked genes (Wutz, 2011; Conrad *et al*, 2012). It is therefore apparent that the production of non-coding RNAs that recruit chromatin modifiers is a common feature of nucleation sites. Non-coding RNAs are well suited for the role as 'connectors' between the genome and chromatin modifiers since they can use their innate ability to base pair to recognize specific RNA or DNA sequences.

### Centromeric heterochromatin is assembled on multiple, redundant nucleation sites

Our analyses reveal that a siRNA-rich nucleation site is modular such that two *dg* fragments (L5 and L8) allow

### 1. Nucleation

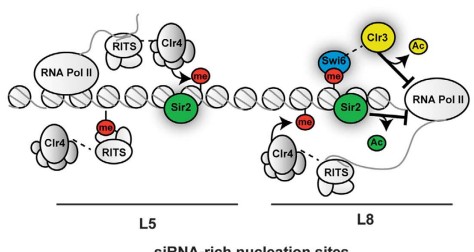

siRNA-rich nucleation sites

### 2. Spreading

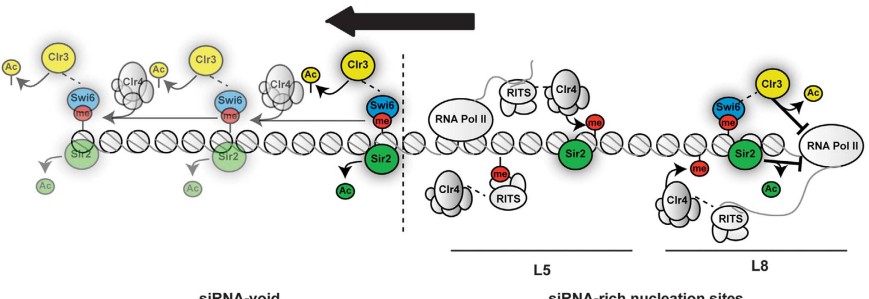

### 3. Maintenance

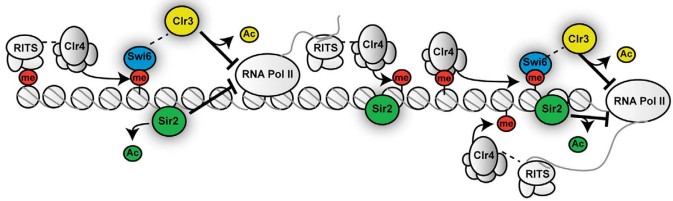

**Figure 7** Model for RNAi and HDACs function in the stepwise assembly of centromeric heterochromatin. H3K9 methylation nucleation. RNAi operates on the siRNA-rich region of the naïve centromeric outer repeat element, where long double-stranded RNAs transcripts are synthesized by RNAPII. At the nucleation sites, the RNAi response generates siRNAs, which guide the RITS complex to homologous nascent transcripts via a base-pairing mechanism. This in turn attracts the chromatin modifier Clr4 to methylate histone H3 on residue lysine 9 (me) on the nucleosomes in the siRNA-rich region. At the L8 nucleation site, HDAC-mediated histone deacetylation is required, in addition to RNAi, to efficiently repress transcription allowing stable H3K9 methylation. H3K9 methylation spreading. Following nucleation, Sir2, Clr3 and Swi6 deacetylate nearby nucleosomes allowing methylation of histone H3 by Clr4. Iterative cycles of deacetylation and methylation result in heterochromatin spreading along the chromatin fibre from the nucleation site. H3K9 methylation maintenance. Combined RNAi and Sir2 actions maintain the H3K9 methylation state at the centromeric repeats over generations. The Clr3 and Sir2 HDACs reduce transcription and consequently histone turnover cooperating with the RNAi pathway to propagate centromeric hetero-chromatin.

*de novo* heterochromatin assembly on minichromosomes. Interestingly, nucleation by the L5 subfragment depends on RNAi components, but does not require Sir2, Clr3 or Swi6$^{HP1}$. In contrast, the seeding of H3K9 methylation by the L8 subfragment requires Sir2, Clr3 and Swi6$^{HP1}$ in addition to active RNAi. Why do L5 and L8 differ in their requirements to nucleate heterochromatin? RNAPII associates with both L5 and L8, and non-coding RNAs are produced from within both segments (Djupedal *et al*, 2005) and this study). However, while a single transcription start site was identified at the L8 nucleation site (Djupedal *et al*, 2005), we detected several transcription start sites originating within the L5 nucleation site suggesting that L5 and L8 contain different types of promoters. We propose that RNAi-mediated recruitment of Clr4 histone methyltranferase is sufficient to repress transcription originating from the L5 element allowing

H3K9me nucleosomes to stably associate with its sequences. Thus, a small region of heterochromatin is formed even in the absence of Sir2. In contrast, HDAC-mediated histone deacetylation would be required, in addition to RNAi, to efficiently repress transcription originating from the L8 nucleation site. In this case, when Sir2 is absent, the high transcriptional activity and associated elevated rate of histone turnover may prevent Clr4 from stably methylating H3K9 (Figure 7A). In support of this model, Clr3 has been shown to contribute to the transcriptional repression of centromeric non-coding transcripts (Sugiyama *et al*, 2007).

Assembly of specialized chromatin domains often depend on multiple nucleation sites (Straub and Becker, 2011). Such redundancy presumably provides general backup mechanisms ensuring the assembly of such large chromatin domains.

### The role of Sir2 in extending heterochromatin domains over entire centromeric repeats

It has been shown previously that the RNAi machinery is required for heterochromatin maintenance on marker genes inserted into centromeric repeats suggesting that RNAi activity is required for heterochromatin spreading on transcriptionally active chromatin (Sadaie *et al*, 2004). In this study, we demonstrated that Sir2, Clr3 and Swi6[HP1] mediate heterochromatin spreading from nucleation sites into flanking endogenous sequences.

Interestingly, in *Saccharomyces cerevisiae*, Sir2 is also utilized to spread a distinct type of repressive chromatin (Rusche *et al*, 2003). However, *S. cerevisiae* completely lacks RNAi, H3K9 methylation and its ligands, the HP1-related proteins. In this system, the Sir2/Sir3/Sir4 silencing complex is recruited to nucleation sites by DNA-bound proteins. Sir2 deacetylates lysine 16 on H4 of nearby nucleosomes to create high-affinity binding sites for Sir3 (Armache *et al*, 2011). This then allows the Sir2/Sir3/Sir4 complex to spread outwards over neighbouring chromatin (Rusche *et al*, 2003). Similar to what has been observed in *S. cerevisiae*, it is likely that the mechanism that spreads H3K9 methylation from siRNA-rich nucleation sites into siRNA-void regions is a self-enforcing process: at nucleation sites, methylation of H3K9 creates binding sites for the chromodomain proteins (such as Swi6 and Chp2) allowing recruitment of the HDAC Clr3 to centromeric repeats (Sugiyama *et al*, 2007; Sadaie *et al*, 2008; Fischer *et al*, 2009). Clr3 would then cooperate with chromatin-bound Sir2 to deacetylate nearby nucleosomes allowing methylation of histone H3 by Clr4. Iterative cycles of this deacetylation and methylation would result in heterochromatin spreading along the chromatin fibre (Figure 7A).

### The maintenance of H3K9 methylation in the absence of RNAi depends on HDACs and Swi6[HP1]

The maintenance of distinctive chromatin domains is known to occur even in the absence of initiating events or their nucleation site (Bonasio *et al*, 2010). Such propagation represents the essence of an epigenetically regulated chromatin domain. In many systems, the inheritance of a chromatin state is dependent on DNA methylation where a replication-coupled mechanism allows recognition of hemi-methylated DNA and methylation of the newly synthesized DNA strand at the replication fork (Kundu and Peterson, 2009). It remains less clear how a particular chromatin state can be propagated in the absence of DNA methylation. Our analyses provide insights into the mechanism that allows the propagation of the heterochromatin state in fission yeast, an organism that lacks DNA methylation.

Fission yeast centromeric heterochromatin is partially disrupted during S phase and RNAi allows the re-establishment of heterochromatin domains following each round of replication. It has been proposed that this cyclical disassembly of heterochromatin allows transient transcription, the generation of new siRNAs and the subsequent recruitment of Clr4 in S phase (Chen *et al*, 2008; Kloc *et al*, 2008). If this was the only mechanism for retaining heterochromatin, in RNAi-deficient cells, H3K9me levels should dramatically decline within a few divisions due to the progressive dilution of pre-existing H3K9 methylated nucleosomes (<1% of wt levels in seven divisions). However, as this

and other studies have shown, H3K9 methylation remains at centromeres in the absence of RNAi (*dcr1Δ*, *ago1Δ*; (Sadaie *et al*, 2004; Partridge *et al*, 2007; Halic and Moazed, 2010; Shanker *et al*, 2010; Reyes-Turcu *et al*, 2011). Thus, a parallel pathway must operate to maintain and propagate H3K9 methylation when RNAi is ablated. Indeed, alternative RNAi-independent pathways that act to maintain heterochromatin at the mating-type locus and telomeres have been identified (Jia *et al*, 2004; Kim *et al*, 2004; Kanoh *et al*, 2005). At centromeres, it has been suggested that Ago1 is critical to propagate H3K9 methylation using Dicer-independent centromeric small RNAs (primal small RNAs or priRNAs) to recruit Clr4 in the absence of RNAi (*dcr1Δ*) (Halic and Moazed, 2010). However, we and others have shown that H3K9 methylation levels are similar in *ago1Δ* and *dcr1Δ* cells, and even *ago1Δdcr1Δ* double mutants (Shanker *et al*, 2010; Reyes-Turcu *et al*, 2011; this study). This indicates that priRNAs play a marginal role in maintaining centromeric heterochromatin.

Here, we provide an alternative mechanism for the RNAi-independent propagation of H3K9 methylation at centromeres (Figure 7B). Our analyses show that both Sir2 and Swi6[HP1] act to maintain H3K9 methylation at centromeres in *dcr1Δ* cell.

Moreover, we demonstrate that the artificial recruitment of Sir2 HDAC activity adjacent to an siRNA-rich nucleating fragment (L5) allows heterochromatin maintenance in the absence of RNAi. This strongly supports the conclusion that Sir2 acts in parallel to RNAi as a maintenance factor for centromeric heterochromatin.

We propose that following the establishment of a centromeric heterochromatin domain, the HDACs Sir2 and Clr3 repress the transcriptional activity of centromeric promoters by deacetylating histone H3 on lysine 9 and 14. This results in reduced histone turnover and in the ability to maintain H3K9 methylation in the absence of RNAi. Importantly, we find that H3K9 methylation cannot be maintained at centromeres in *sir2Δclr3Δ* double mutants, even though they retain active RNAi. We surmise that in *sir2Δclr3Δ* cells higher levels of centromeric transcription causes elevated rates of histone turnover preventing the stable methylation of H3K9 on resident nucleosomes by Clr4.

Other analyses indicate that defective nuclear exosome function (*rrp6Δ*) also results in loss of H3K9me2 from centromeric repeats in the absence of RNAi (Reyes-Turcu *et al*, 2011). Cells lacking both Sir2 and Rrp6 have reduced H3K9me2, but in contrast to *sir2Δclr3Δ* cells, it is not abolished (Supplementary Figure S5E). This observation raises the possibility that Sir2 and Rrp6 act together to maintain H3K9me2 but this requires further investigation to tease out their relationship.

The analyses presented provide insight into how a distinct chromatin domain is established, extended and propagated. The identification of Sir2 as a heterochromatin maintenance factor in a system that lacks DNA methylation raises the possibility that Sirtuins in other organisms also contribute to the propagation of specialized chromatin domains. Moreover, our approach demonstrates that the comparison of the histone-modification patterns across chromosomal domains using both establishment and maintenance assays will be required to completely decipher the epigenomes of metazoa.

## Materials and methods

### Yeast strains, plasmids and standard techniques

For fission yeast strains, see Supplementary Table S1. Standard procedures were used for bacterial and fission yeast growth, genetics and manipulations (Moreno *et al*, 1991). Strains containing minichromosomes were grown in PMG medium (Pombe Minimal Glutamate medium) lacking adenine and uracil, otherwise strains were grown in YES medium (yeast extract with supplements). Serial (1:5) dilutions of cells were spotted onto YES medium containing low adenine, full adenine with DMSO or TBZ 10 µg/ml. Cells were grown at 25°C for 5 days. Gene deletions and tagging were carried out by lithium acetate transformation method (Moreno *et al*, 1991). Selections were performed on PMG with according auxotrophy or on YES with appropriate antibiotic at 32°C. The *cid12-ha* hypomorphic allele was constructed using a PCR-based module method and contains a $3 \times HA$ (haemagglutinin) moiety at the carboxyl terminus. The ectopic L5 silencing system was modified (Wheeler *et al*, 2009) by cloning the L5 fragment with *SpeI/ClaI* into BW5/6-4TetO plasmid, upstream of *4* of the *dg TetO-ade6$^+$* (described in (Bayne *et al*, 2010) to generate the *L5-4TetO-ade6$^+$* reporter. *PstI*-digested plasmid *BW5/6-L5-4TetO* was integrated at *ura4$^+$*. For pDUAL-TetR$^{off}$-$2 \times$ FLAG-Sir2, *sir2$^+$* was cloned as described previously for the *stc1$^+$* gene (Bayne *et al*, 2010).

### Sir2 antibody production

Recombinant Sir2 fragment (amino acids 1–113) fused to GST was injected into rabbits. The antibodies, obtained following three injections, were affinity purified on nitrocellulose membrane and eluted with glycine.

### Minichromosome cloning, selection system and stability

The following minichromosomes were used in this study: (i) in Figure 1: MC-*dg* (pcc2K″; (Baum *et al*, 1994) contains a 5.6 kb outer repeat sequence corresponding to the *dg* element; MC-L6 (pLCC2) contains a 3.2 kb of *dg* siRNA-rich fragment; MC-L7 (pLCC1) contains a 2.2 kb *dg* siRNA-void fragment; MC-L5 (pLCC3-Fragment A) contains 1.6 kb of the *dg* element; MC-L8 (pLCC7-Fragment E) contains 1.6 kb of the *dg* element; and MC-L9 (pLCC9-Fragment J) contains 0.6 kb of the *dg* fragment. To clone MC-L5, MC-L6, MC-L7; MC-L8 and MC-L9, different dg fragments were amplified with primers (Supplementary Table S2) bearing *Bam*HI and *Nco*I sites and cloned into pcc2K″ digested with the same enzymes. (ii) In Figure 2: MC-*dg″* (pHHcc2; (Baum *et al*, 1994) contains two tandemly repeated 5.6 kb outer repeat sequence corresponding to the *dg* element. All minichromosomes used contain, in addition to full-length *dg* element or *dg* fragments, the fission yeast centromeric central domain DNA (cc) and the *ura4$^+$* and *sup3-5* (suppressor of *ade6-704)* selection systems. Cells without *ura4$^+$* cannot grow on–uracil plates, while *ade6-704* cells do not grow without adenine and form red colonies on 1/10th adenine plates. The *sup3-5-tRNA* gene suppresses a premature stop codon in *ade6-704,* allowing growth on –adenine plates. Minichromosomes were introduced into *S. pombe* by electroporation and transformants were selected by growth on PMG–ura–ade at 32°C for 5–7 days. For quantification Figure 1C: primary transformants were replica-plated from PMG–ura–ade plates into YES low ade plates. The number of white colonies (containing mitotically stable minichromosomes) was counted and expressed as percentage of the total number of colonies. To confirm that plasmids were behaving episomally and had not integrated, cells (100–1000) were plated onto YES 1/10 adenine and allowed to form colonies. Wt strains containing plasmids typically exhibit 80–90% of white/sectored colonies and samples exhibiting <2% of integrations (i.e., white colonies in the mutants) were included in the quantification. For quantification Figure 4E: number of white-sectored colonies (containing episomal minichromosomes) were counted and expressed as percentage of the total number of colonies. Completely white colonies were not included in the quantification because they contain integrated minichromosomes. Specific strains and primer pairs were used to distinguish *dg* sequences on plasmids from those at endogenous centromeres. Primers across the insertion site in the plasmid only detect the *dg* of the minichromosomes.

### Chromatin immunoprecipitation

Cells were grown at 32°C either in YE-rich media. Primary transformants containing minichromosomes were grown in PMG–ura–ade liquid media. To confirm that plasmids were behaving episomally and had not integrated, a plasmid stability test was performed at the time of fixation. Cells (100–1000) were plated onto YES 1/10 adenine and allowed to form colonies. Samples exhibiting no integrations were used for ChIP.

ChIP was performed essentially as described (Bayne *et al*, 2010). Briefly, for H3K9me2 ChIP, cells were fixed with 1% PFA for 15 min at room temperature. Cells were lysed using a bead beater (Biospec Products) and sonicated using a Bioruptor (Diagenode) sonicator for a total of 15 min (30 s ON and OFF cycle). One microlitre of H3K9me2 antibody (m5.1.1, (Nakagawachi *et al*, 2003)); 5 µl of GFP antiserum (Molecular Probes); 5 µl of RNAPII 8WG16 antibody (Covance, MMS-126R); and 1 µl of FLAG antibody (Sigma) were used for IPs. For Swi6 ChIP, cells were fixed for 30 min at 18°C after a 2 h shift at 18°C. Three microlitre of Swi6 rabbit polyclonal antibody (Thermo Scientific:Ab PA1-4977) was used for IP.

### PCR reactions

Primers used are listed in Supplementary Table S2. Real-time PCR was performed in the presence of SYBR Green on a Roche LightCycler. Data were analysed with LightCycler 480 Software 1.5.0.39. Relative enrichments were calculated as the ratio of product of interest to control product (*act1$^+$*) in IP over input, expressed as percentage of wt. Histograms represent data from three biological replicates. Error bars: s.d.'s of three biological replicates.

### RNA analysis

RT–PCR and 5′ RACE–PCR were performed as previously described (Choi *et al*, 2011). Northern analysis of centromeric siRNAs and long non-coding CEN transcripts were performed as described (Bayne *et al*, 2010; Buscaino *et al*, 2012).

### Cytology

Immunolocalization was performed as described previously (Bayne *et al*, 2010). Cells were fixed with 3.7% PFA for 10 min, plus 0.05% glutaraldehyde for tubulin staining. Antibodies used were TAT1 anti-tubulin 1:15 (K. Gull), anti-CENP-A$^{Cnp1}$ 1:1000 and anti-GFP 1:200 (Molecular Probes); anti-Sir2 1:50 Alexa Fluor 594- and 488-coupled secondary antibodies were used at 1:1000 (Invitrogen).

### Supplementary data

Supplementary data are available at *The EMBO Journal* Online (http://www.embojournal.org).

## Acknowledgements

We thank HD Folco, A Kagansky, SA White, S Catania and L Subramanian for input and reagents. We are grateful to T Urano for the anti-H3K9me2 antibody, K Scott, D Moazed and S Grewal for strains and materials. EL was supported by EC FP7 Marie Curie Fellowship PIEF-GA-2009-235892; PA is supported by the Wellcome Trust 4 Year PhD programme in Cell Biology (093852/Z/10/Z). The Centre for Cell Biology is supported by core funding from the Wellcome Trust (092076/Z/10/Z). RCA is a Wellcome Trust Principal Research Fellow and this research was supported by the Wellcome Trust. (065061/Z/01/A and 095021/Z/10/Z).

*Author contributions*: EL, AB, AP and RCA conceived and designed the experiments. EL and AB performed most of the experiments. PA, AP and GH performed minichromosome analyses. EL, AB and RCA wrote the manuscript.

## Conflict of interest

The authors declare that they have no conflict of interest.

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
