## [Review Process File · The EMBO Journal]

Manuscript EMBO-2012-84216

Distinct roles for Sir2 and RNAi in centromeric heterochromatin nucleation, spreading & maintenance

Alessia Buscaino, Erwan Lejeune, Pauline Audergon, Georgina Hamilton, Alison Pidoux and Robin C. Allshire

Corresponding author: Robin C. Allshire, The Wellcome Trust

Review timeline:

Submission date:	13 December 2012
Preliminary Decision:	30 January 2013
Editorial Decision:	04 February 2013
Revision received:	27 February 2013
Editorial Decision:	04 March 2013
Revision received:	05 March 2013
Accepted:	08 March 2013

Transaction Report:

Editor: Anke Sparmann

Preliminary Decision

30 January 2013

Thank you for submitting your manuscript (EMBOJ-2012-84216) to our editorial office. Please find enclosed the comments of two of the three reviewers whom we had asked to evaluate your research for The EMBO Journal. Unfortunately, we are still waiting for a considerably delayed third report that the referee promised to deliver very shortly. Still, given the comments I have at hand, I can make a preliminary decision now. This decision is still subject to change in the unlikely event that the third referee offers strong and convincing reasons for doing so. However, as you can see, both current reviewers appreciate your study and are in general supportive of publication in The EMBO Journal. Nevertheless, they do raise a number of concerns, which will require some additional experimentation. Therefore, I would like to give you the opportunity to already start making the requested changes and additions to the manuscript that would render the paper suitable for publication in the view of these two reviewers. I will forward the comments of the third referee to you as soon as we receive them, together with our final editorial decision.

My apologies for the delay in coming to a final conclusion.

 REFEREE COMMENTS

Referee #1

Heterochromatin is a condensed chromatin structure and involved in various genome functions such as epigenetic repression of gene expression and centromere function. The fission yeast provides a good model system for investigating heterochromatin formation. In fission yeast, heterochromatin, which is defined by methylation of histone H3K9 (H3K9me) and its binding partner Swi6/HP1, is preferentially enriched across large chromosomal domains at the pericentromeres, subtelomeres and the mating-type locus. Assembly of large heterochromatin domain is thought to require three events: establishment, spreading and maintenance. RNAi-dependent system is shown to establish heterochromatin at pericentromere, but the molecular mechanism for spreading and maintenance is still obscure. In this manuscript, authors showed that histone deacetylases Sir2 and Clr3 and the Swi6/HP1 are required for spreading from nucleation sites and the maintenance of heterochromatin. In addition, they discovered two distinct RNAi-directed systems, Sir2-dependent and Sir2-independent systems, function for establishment of heterochromatin. The findings are informative to understand the general mechanism of assembly of large chromatin domain and will attract interests of broad range of readers. They used ingenious systems such as mini-chromosome, ectopic heterochromatin system, artificial tethering of Sir2, to show the data that support their proposal and the experiments are generally well designed. However, there are several important issues to clarify before publishing in EMBO Journal, which are listed below.

Major points

1. Authors claim that RNAPII activity defines redundant nucleation sites within the siRNA-rich region in Fig 1. This conclusion mainly depends on the comparison of the results using the MC-L8 and MC-L9, the former contains P1 promoter and the latter does not. Since MC-L9 only retains upstream region of P1 promoter, it is possible that cis-element that is deleted in MC-L9, not the promoter or RNAPII activity, contributes heterochromatin assembly. To claim the importance of RNAPII activity, authors should analyze the presence of RNAPII and transcripts in MC-L8 and MC-L9, like they showed in Fig 1E and F for MC-L5.
2. Authors indicate that Sir2, Clr3 and Swi6/HP1 are crucial for spreading of heterochromatin into siRNA-void region. I agree with their conclusion. However, authors' group and other groups showed that heterochromatin spreads into the marker genes inserted in pericentromeric heterochromatin in an RNAi-dependent manner. As a result, the silencing and H3K9me on the marker genes are abolished in RNAi mutants, in which substantial heterochromatin is still kept at the pericentromere. In contrast, deletion of sir2 does not affect the silencing (Fig 4A), indicating Sir2 is dispensable for spreading into the marker genes. Therefore, I think there are two distinct spreading systems, Sir2-dependent and RNAi-dependent systems. I suspect that Sir2-systems functions for spreading into transcriptionally inactive region, while RNAi-dependent system functions for spreading into transcriptionally active region. In any case, authors should discuss this point in the manuscript.
3. Authors clearly indicate that Sir2, Clr3 and Swi6/HP1 are required for maintenance of heterochromatin independently on RNAi. A recent report showed that Rrp6, a component of nuclear exosome, plays a role in the maintenance of heterochromatin (Reyes-Turcu et al. Nat. Struct. Mol. Biol. 2011). Like sir2Δdcr1Δ double mutants, rrp6Δago1Δ double mutant loses H3K9me. It looks like that rrp6 is also involved in sir2 system. Authors should clarify this point.
4. Authors propose that Sir2 and Clr3 independently suppress transcription from centromeric repeats and that, in the absence of both HDACs, high transcriptional activity prevents H3K9 methylation propagation even in the presence of active RNAi (p14). To propose this, it is important to show the sir2clr3 double mutant shows higher transcriptional activity than each single mutant.
5. Based on the results in Fig 6, authors claim that Sir2 is sufficient to "maintain" heterochromatin in the absence of RNAi. L5 cannot induce heterochromatin at ectopic loci in the absence of RNAi, but artificial tethering of Sir2 to L5 rescue the defects in heterochromatin formation. I think this result indicates that the Sir2 induces heterochromatin formation in the absence RNAi. In this context, I think authors should show the results of Sir2-tethering in the absence of L5.

Minor points

1. Authors should show schematics of mini-chromosome for better understanding of the experiments, though they describe the detail of the structure in Materials and Methods.
2. In Fig 5B, Swi6 is significantly decreased in *clr3Δ*. However, another paper reported the Swi6 is not decreased in *clr3Δ* mutants (Shimada et al. Genes Dev. 2009). What causes the difference?

Referee #3

The heterochromatin-regulating pathways, including RNAi and enzymes for different histone modifications, are interconnected and with certain redundancy. In this study, Lejeune et al. started from the observation that although K9me2, the heterochromatin mark deposited by Clr4, spans the whole pericentromeric region in *S. pombe*, there are certain loci enriched for siRNA ("siRNA-rich") and lack of it ("siRNA-void"). Using a mini-chromosome system, they found that siRNA-rich regions can serve as nucleation sites for heterochromatin while the other cannot. Further analysis of the "siRNA-rich" region reveals two types of nucleation sites: those that require RNAi but not the structural protein Swi6 or the histone deacetylases Sir2, and those that require all. Once heterochromatin is established on nucleation sites, its subsequent spreading into "siRNA-void" regions not only requires Swi6 but also histone deacetylases Clr3 and Sir2. In summary, Lejeune et al. deciphered the role of each player at different stages of heterochromatin establishment and maintenance in *S. pombe*. And the principle of these roles is conserved from fungi to higher eukaryotes, thus should appeal to a broad interest.

In *S. cerevisiae* where RNAi does not exist, Sir2 and its deacetylases activity became essential to heterochromatin. Here, its function in *S. pombe* is clearly demonstrated. Besides its distinct role from RNAi and Clr3 at peri-centromeric heterochromatin, it can establish heterochromatin ectopically even in the absence of RNAi when tethered to the locus, again showing the evolutionary conservation.

Yet, the manuscript can be further improved if the following questions are addressed:

1. The authors identified two different types of nucleation sites, L8 requiring more repression mechanism than L5.
 - 1) Is Clr3 also required at L8?
 - 2) Why is the difference? Is there evidence of higher Pol II activity at L8? If so, a similar example can be found in *S. cerevisiae*. One of the two cryptic mating type loci, HMR contains a stronger promoter and requires an extra cell cycle event along with Sir proteins for heterochromatin establishment, while the other locus HML with a weaker promoter only requires the latter.
2. Fig 5, is p-dg in "siRNA-void" region? If so, how about the "siRNA-rich region"?
3. Fig 6, is Sir2 recruited to the ectopic heterochromatin if not artificially tethered? Or can ectopic heterochromatin be established in the presence of RNAi but without Sir2?

Minor points:

1. Table S2, BWP100 is for figure 6C instead of 5C.
2. Please also provide the sequences of primers used to amplify A-F in Fig 3A. It could also be useful to list the coordinates of the L5-L9 fragments.

Editorial Decision

4 February 2013

We have now received the final referee report, which you will find enclosed below. As is apparent, also reviewer #2 is supportive of publication in The EMBO Journal. Nevertheless, s/he requests a

clarification of the experiment shown in Figure 6B&C and suggests several minor modifications. Given the positive comments provided, I would like to invite you to submit a suitably revised manuscript to The EMBO Journal. I should add that it is our policy to allow only a single major round of revision and that it is therefore important to address the raised concerns at this stage.

When preparing your letter of response to the referees' comments, please bear in mind that this will form part of the Review Process File, and will therefore be available online to the community. For more details on our Transparent Editorial Process, please visit our website: <http://www.nature.com/emboj/about/process.html>

Thank you for the opportunity to consider your work for publication.
I look forward to your revision!

REFEREE COMMENTS

Referee #2

This is a very interesting paper. The Allshire group has been responsible for many of the most insightful papers on the function of the *S. pombe* centromere and the role of RNAi in controlling the function of the centromere and its heterochromatic properties. This paper is another important contribution in that lineage. In this work, they address the long enigmatic role of the Sir2 protein in the silencing of genes in centromeres or near centromere repeats, and in the function of centromeres. As a non-*pombe* investigator, Sir2 has long seemed to me to be equivalent to the eccentric uncle whose existence is acknowledged but then promptly ignored. A few papers have identified its role in epigenetic processes in *pombe*, but the weakness of its phenotypes has led to its relegation as a supporting actor rather than a main player.

In this work, by applying the most modern analytical tools, infused with elegant genetic studies, the authors have identified roles of Sir2 and another deacetylase Clr3 in the spreading of heterochromatin from RNAi-dependent sites of nucleation in the centromere repeats. The functions of the two deacetylases is partially overlapping, accounting for the weaker phenotypes of the individual mutants. (I take exception to the claim of redundancy as true redundancy is impossible except in the case of extremely recent duplications in any Darwinian biology.)

All in all, this is a well executed contribution to the literature in this field and should be promptly published.

From my perspective, peer review has fallen into a rut in which reviewers are never satisfied with the extent of the data in a paper, and feel entitled, after reflecting on the manuscript for a few hours, do demand months of additional work. This is nuts. However, I do find one experiment to either be unclear, or in need of a clarifying additional experiment, which I suspect the authors have already done. I refer to the very interesting experiment in figures 6B and C in which the ability of a tethered Sir2 is evaluated for its ability to silence a reporter gene. Here is my puzzle: In the absence of the Tet repressor Sir2 fusion, a fraction of cells have silenced the reporter in a fashion that is completely dependent on DCR. In contrast, the tethered Sir2 is also able to silence the reporter in a fashion that is independent of DCR, yet that silencing is quantitatively identical to that seen in the wild-type cell. Unless the latter experiment is done in a strain lacking a wild-type Sir2 (possible but unclear to me), it seems to me that there should be an additive effect. If there isn't, then something odd is happening that would make me want to see this experiment repeated in a strain in which the tethered Sir2 is independent of any centromeric repeat.

The model slide seems to fall short of its potential in that it lacks any representation of kinetics....which steps are first and which follow? Modest improvement of the model slide would enhance its impact.

Minor points

The Wiren references is duplicated in the literature cited.

The Bonasio and Reinberg review is recent and is cited numerous times, yet I cannot think of a review more lacking in original ideas than that one. Surely the authors can do a little better than that.

Page 12 line 16 (approximately) the word "than" should be deleted

Page 17, line 11 the word "associates" should be deleted.

1st Revision - authors' response

27 February 2013

We thank the reviewers for their positive and helpful comments that improved the quality of our manuscript. Our point-by-point response is listed below.

Referee #1

Heterochromatin is a condensed chromatin structure and involved in various genome functions such as epigenetic repression of gene expression and centromere function. The fission yeast provides a good model system for investigating heterochromatin formation. In fission yeast, heterochromatin, which is defined by methylation of histone H3K9 (H3K9me) and its binding partner Swi6/HP1, is preferentially enriched across large chromosomal domains at the pericentromeres, subtelomeres and the mating-type locus. Assembly of large heterochromatin domain is thought to require three events: establishment, spreading and maintenance. RNAi-dependent system is shown to establish heterochromatin at pericentromere, but the molecular mechanism for spreading and maintenance is still obscure. In this manuscript, authors showed that histone deacetylases Sir2 and Clr3 and the Swi6/HP1 are required for spreading from nucleation sites and the maintenance of heterochromatin. In addition, they discovered two distinct RNAi-directed systems, Sir2-dependent and Sir2-independent systems, function for establishment of heterochromatin. The findings are informative to understand the general mechanism of assembly of large chromatin domain and will attract interests of broad range of readers. They used ingenious systems such as mini-chromosome, ectopic heterochromatin system, artificial tethering of Sir2, to show the data that support their proposal and the experiments are generally well designed. However, there are several important issues to clarify before publishing in EMBO Journal, which are listed below.

Major points

1. Authors claim that RNAPII activity defines redundant nucleation sites within the siRNA-rich region in Fig 1. This conclusion mainly depends on the comparison of the results using the MC-L8 and MC-L9, the former contains P1 promoter and the latter does not. Since MC-L9 only retains upstream region of P1 promoter, it is possible that cis-element that is deleted in MC-L9, not the promoter or RNAPII activity, contributes heterochromatin assembly. To claim the importance of RNAPII activity, authors should analyze the presence of RNAPII and transcripts in MC-L8 and MC-L9, like they showed in Fig 1E and F for MC-L5.

Response: We appreciate the concerns of the reviewer and have performed the experiments suggested. As shown in the new Figure 1 E and F, RNA polymerase II associates with the L8 but not with the L9 fragment. In addition, RT-PCR analyses detect transcripts originating from the L8 but not from the L9 fragment.

2. Authors indicate that Sir2, Clr3 and Swi6/HP1 are crucial for spreading of heterochromatin into siRNA-void region. I agree with their conclusion. However, authors' group and other groups showed that heterochromatin spreads into the marker genes inserted in pericentromeric heterochromatin in an RNAi-dependent manner. As a result, the silencing and H3K9me on the marker genes are abolished in RNAi mutants, in which substantial heterochromatin is still kept at the pericentromere. In contrast, deletion of sir2 does not affect the silencing (Fig 4A), indicating Sir2 is dispensable for spreading into the marker genes. Therefore, I think there are two distinct spreading systems, Sir2-dependent and RNAi-dependent systems. I suspect that Sir2-systems functions for spreading into transcriptionally inactive region, while RNAi-dependent system functions for spreading into transcriptionally active region. In any case, authors should discuss this point in the manuscript.

Response: We thank the referee for the suggestion. We now discuss this possibility (p18).

3. Authors clearly indicate that Sir2, Clr3 and Swi6/HP1 are required for maintenance of

heterochromatin independently on RNAi. A recent report showed that Rrp6, a component of nuclear exosome, plays a role in the maintenance of heterochromatin (Reyes-Turcu et al. Nat. Struct. Mol. Biol. 2011). Like sir2Δdcr1Δ double mutants, rrp6Δago1Δ double mutant loses H3K9me. It looks like that rrp6 is also involved in sir2 system. Authors should clarify this point.

Response: We have now analysed H3K9 methylation level in rrp6Δsir2Δ double mutant. Substantial amount of H3K9 methylation was still detected in this mutant background suggesting that Rrp6 is not involved in the Sir2 system. For this reason, we have decided not to include this data in the manuscript.

4. Authors propose that Sir2 and Clr3 independently suppress transcription from centromeric repeats and that, in the absence of both HDACs, high transcriptional activity prevents H3K9 methylation propagation even in the presence of active RNAi (p14). To propose this, it is important to show the sir2clr3 double mutant shows higher transcriptional activity than each single mutant.

Response: The referee has raised a critical point and we are glad that these experiments have been suggested. We have now extended our analyses of the sir2Δclr3Δ double mutant. Our new results strongly support the hypothesis that “*in the absence of both HDACs, high transcriptional activity prevents H3K9 methylation propagation even in the presence of active RNAi*” as stated in p14.

In detail, we show that

1. Marker gene silencing is alleviated in sir2Δclr3Δ double mutant (new Figure 5A)
2. sir2Δclr3Δ cells display increased sensitivity to TBZ (new Figure 5B).
3. High levels of siRNAs are detected in sir2Δclr3Δ cells demonstrating that RNAi is still active (new Figure 5C).
4. Long unprocessed CEN transcripts accumulate in sir2Δclr3Δ but not in the corresponding single mutants (new Figure 5E).

5. Based on the results in Fig 6, authors claim that Sir2 is sufficient to "maintain" heterochromatin in the absence of RNAi. L5 cannot induce heterochromatin at ectopic loci in the absence of RNAi, but artificial tethering of Sir2 to L5 rescue the defects in heterochromatin formation. I think this result indicates that the Sir2 induces heterochromatin formation in the absence RNAi. In this context, I think authors should show the results of Sir2-tethering in the absence of L5.

Response: The experiments in Figure6 demonstrate that once established Sir2 can maintain heterochromatin even in the absence of RNAi. We did not anticipate that tethering of Sir2 would be sufficient to establish heterochromatin *de novo*. This is because the experiments of Figure1 point to a crucial role for RNAi in establishing heterochromatin. However, we have now performed the important control suggested by all three reviewers (new Figure S5 C and D). As predicted, this analysis reveals that tethering of Sir2 is not sufficient to assemble heterochromatin at an ectopic locus, in the absence of the L5 nucleation site. These results confirm that RNAi activity is necessary to establish heterochromatin *de novo*. Once established, Sir2 can maintain heterochromatin in the absence of RNAi (Figure 6).

Minor points

1. Authors should show schematics of mini-chromosome for better understanding of the experiments, though they describe the detail of the structure in Materials and Methods.

Response: This has been added in Figure S2C.

2. In Fig 5B, Swi6 is significantly decreased in clr3Δ. However, another paper reported the Swi6 is not decreased in clr3Δ mutants (Shimada et al. Genes Dev. 2009). What causes the difference?

Response: The reviewer is referring to an experiment using in gel semi-quantitative PCR. In our own experience, this type of analyses often does not measure accurately the amount of protein associated with a given locus. This could explain the apparent discrepancy between our quantitative data and the data published by (Shimada et al. 2009).

Referee #2

*This is a very interesting paper. The Allshire group has been responsible for many of the most insightful papers on the function of the *S. pombe* centromere and the role of RNAi in controlling the function of the centromere and its heterochromatic properties. This paper is another important contribution in that lineage. In this work, they address the long enigmatic role of the Sir2 protein in the silencing of genes in centromeres or near centromere repeats, and in the function of centromeres. As a non-*pombe* investigator, Sir2 has long seemed to me to be equivalent to the eccentric uncle whose existence is acknowledged but then promptly ignored. A few papers have identified its role in epigenetic processes in *pombe*, but the weakness of its phenotypes has led to its relegation as a supporting actor rather than a main player.*

In this work, by applying the most modern analytical tools, infused with elegant genetic studies, the authors have identified roles of Sir2 and another deacetylase Clr3 in the spreading of heterochromatin from RNAi-dependent sites of nucleation in the centromere repeats. The functions of the two deacetylases is partially overlapping, accounting for the weaker phenotypes of the individual mutants. (I take exception to the claim of redundancy as true redundancy is impossible except in the case of extremely recent duplications in any Darwinian biology.)

All in all, this is a well executed contribution to the literature in this field and should be promptly published.

From my perspective, peer review has fallen into a rut in which reviewers are never satisfied with the extent of the data in a paper, and feel entitled, after reflecting on the manuscript for a few hours, do demand months of additional work. This is nuts. However, I do find one experiment to either be unclear, or in need of a clarifying additional experiment, which I suspect the authors have already done. I refer to the very interesting experiment in figures 6B and C in which the ability of a tethered Sir2 is evaluated for its ability to silence a reporter gene. Here is my puzzle: In the absence of the Tet repressor Sir2 fusion, a fraction of cells have silenced the reporter in a fashion that is completely dependent on DCR. In contrast, the tethered Sir2 is also able to silence the reporter in a fashion that is independent of DCR, yet that silencing is quantitatively identical to that seen in the wild-type cell. Unless the latter experiment is done in a strain lacking a wild-type Sir2 (possible but unclear to me), it seems to me that there should be an additive effect. If there isn't, then something odd is happening that would make me want to see this experiment repeated in a strain in which the tethered Sir2 is independent of any centromeric repeat.

Response:

We thank the reviewer for the positive comments. The experiments in Figure 6 demonstrate that once established Sir2 can maintain heterochromatin even in the absence of RNAi and not that tethering of Sir2 is sufficient to nucleate heterochromatin *de novo*. Actually, we did not anticipate that tethering of Sir2 would be sufficient to establish heterochromatin. This is because the experiments of Figure 1 point to a crucial role for RNAi in establishing heterochromatin. However, we have now performed the important control suggested by all three reviewers (new Figure S5 C and D). As predicted, this analysis reveals that tethering of Sir2 is not sufficient to assemble heterochromatin at an ectopic locus, in the absence of the L5 nucleation site. These results confirm that RNAi activity is necessary to establish heterochromatin *de novo*. Once established, Sir2 can maintain heterochromatin in the absence of RNAi (Figure 6).

The model slide seems to fall short of its potential in that it lacks any representation of kinetics...which steps are first and which follow? Modest improvement of the model slide would enhance its impact.

Response: We have now modified the model (new Figure 7) taking in consideration the reviewer's suggestions.

Minor points

The Wiren references is duplicated in the literature cited.

Response: This has been corrected.

The Bonasio and Reinberg review is recent and is cited numerous times, yet I cannot think of a

review more lacking in original ideas than that one. Surely the authors can do a little better than that.

Response: We have incorporated additional references.

Page 12 line 16 (approximately) the word "than" should be deleted

Response: This has been corrected.

Page 17, line 11 the word "associates" should be deleted.

Response: This has been corrected.

Referee #3

*The heterochromatin-regulating pathways, including RNAi and enzymes for different histone modifications, are interconnected and with certain redundancy. In this study, Lejeune et al. started from the observation that although K9me2, the heterochromatin mark deposited by Clr4, spans the whole pericentromeric region in *S. pombe*, there are certain loci enriched for siRNA ("siRNA-rich") and lack of it ("siRNA-void"). Using a mini-chromosome system, they found that siRNA-rich regions can serve as nucleation sites for heterochromatin while the other cannot. Further analysis of the "siRNA-rich" region reveals two types of nucleation sites: those that require RNAi but not the structural protein Swi6 or the histone deacetylases Sir2, and those that require all. Once heterochromatin is established on nucleation sites, its subsequent spreading into "siRNA-void" regions not only requires Swi6 but also histone deacetylases Clr3 and Sir2. In summary, Lejeune et al. deciphered the role of each player at different stages of heterochromatin establishment and maintenance in *S. pombe*. And the principle of these roles is conserved from fungi to higher eukaryotes, thus should appeal to a broad interest.*

*In *S. cerevisiae* where RNAi does not exist, Sir2 and its deacetylases activity became essential to heterochromatin. Here, its function in *S. pombe* is clearly demonstrated. Besides its distinct role from RNAi and Clr3 at peri-centromeric heterochromatin, it can establish heterochromatin ectopically even in the absence of RNAi when tethered to the locus, again showing the evolutionary conservation.*

Yet, the manuscript can be further improved if the following questions are addressed:

1. The authors identified two different types of nucleation sites, L8 requiring more repression mechanism than L5.

1) Is Clr3 also required at L8?

Response: Yes, Clr3 is required for heterochromatin establishment on the L8 nucleation site but not on the L5 nucleation site (new Figure 3C).

*2) Why is the difference? Is there evidence of higher Pol II activity at L8? If so, a similar example can be found in *S. cerevisiae*. One of the two cryptic mating type loci, HMR contains a stronger promoter and requires an extra cell cycle event along with Sir proteins for heterochromatin establishment, while the other locus HML with a weaker promoter only requires the latter.*

Response: We have now analysed by quantitative ChIP analyses and quantitative RT-PCR, RNA Pol II occupancy on the L5 and L8 nucleation sites and RNA levels produced from L5 and L8 (new figures 1E, F, G and H). These analyses suggest that L8 does not contain a stronger promoter than L5. However while a single transcription start site was identified at the L8 nucleation site (Djupedal et al. 2005) we detected several transcription start sites originating within the L5 nucleation site (figure S4A). This might reflect the different transcriptional activity of these two promoters: before heterochromatin nucleation, the L8 element could act as a "canonical" RNA polymerase II promoter while the L5 element could act as cryptic promoter. In this scenario, the HDACs Sir2 and Clr3 could

be required to silence the L8 promoter while the RNAi machinery could be sufficient to silence the L5 promoter. These different possibilities are now included in the discussion.

2. Fig 5, is *p-dg* in "siRNA-void" region? If so, how about the "siRNA-rich region"?

Response: We have now analysed the siRNA-rich region of *dg* (new Figure S4C). Also in this region, no K9 methylation is detected in a *sir2Ddcr1D* double mutant.

3. Fig 6, is *Sir2* recruited to the ectopic heterochromatin if not artificially tethered? Or can ectopic heterochromatin be established in the presence of RNAi but without *Sir2*?

Response: We have now performed these experiments. As shown in new Figure S 5A and B, *Sir2* is partially dispensable for the L5-driven heterochromatin establishment. In addition, we also show (new Figure S5C and D) that tethering of *Sir2* is not sufficient to assemble heterochromatin at an ectopic locus, in the absence of the L5 nucleation site. These results are consistent with the results shown in Figure 1 and they confirm that RNAi activity is sufficient and necessary to establish heterochromatin *de novo*. Once established, *Sir2* can maintain heterochromatin in the absence of RNAi (Figure 6).

Minor points:

1. Table S2, *BWP100* is for figure 6C instead of 5C.

Response: This has been corrected.

2. Please also provide the sequences of primers used to amplify A-F in Fig 3A. It could also be useful to list the coordinates of the L5-L9 fragments.

Response: The primers used to amplify region A to F are now listed in Table S2. The coordinates of the L5-L9 fragments are listed in Figure S2.

References:

- Djupedal I, Portoso M, Spahr H, Bonilla C, Gustafsson CM, Allshire RC, Ekwall K. 2005. RNA Pol II subunit Rpb7 promotes centromeric transcription and RNAi-directed chromatin silencing. *Genes Dev* **19**: 2301-2306.
- Shimada A, Dohke K, Sadaie M, Shinmyozu K, Nakayama J, Urano T, Murakami Y. 2009. Phosphorylation of Swi6/HP1 regulates transcriptional gene silencing at heterochromatin. *Genes Dev* **23**: 18-23.

2nd Editorial Decision

04 March 2013

Thank you for submitting your revised manuscript for our consideration. I am pleased to inform you that in light of the re-review comments from one of the original referees (provided below), we are happy to accept the paper, pending modification of a few additional points:

- the referee is not entirely satisfied with your answer to his/her major point 3 and a further clarification of the level of H3K9me in the *rrp6deltasir2delta* double mutant is required.
- Please indicate the number of biological replicates performed to generate the data as well as information about the statistical test used to create the error bars to all Figure legends.
- Please complete and sign the linked license agreements (see below).

I will now return your manuscript to you for one additional round of minor revision. After that we should be able to swiftly proceed with formal acceptance and production of the manuscript!

If you have any questions, please do not hesitate to contact me directly.

REFEREE COMMENTS

Referee #1

The revised manuscript adequately answered all the issues raised by me (referee #1) except one point (major point 3). Authors found that "substantial" amount of H3K9me was detected in *rrp6Δsir2Δ* double mutant and claimed that this result indicates that *sir2* is not involved in *rrp6*-pathway. I do not know what "substantial" means. Is the level of H3K9me observed in the double mutant similar to those of the single mutant? If that is the case, I think that the results genetically indicate that Rrp6 and Sir2 is epistatic and both proteins function in the same pathway for H3K9me. This is an important point for understanding the nature of Sir2-dependent pathway and authors should clarify this point.

2nd Revision - authors' response

05 March 2013

(Editor's comments): Thank you for submitting your revised manuscript for our consideration. I am pleased to inform you that in light of the re-review comments from one of the original referees (provided below), we are happy to accept the paper, pending modification of a few additional points:

*- the referee is not entirely satisfied with your answer to his/her major point 3 and a further clarification of the level of H3K9me in the *rrp6Δsir2Δ* double mutant is required.*

Response: We have now added our H3K9me2 ChIP in Figure S5. In addition we have modified the discussion and added the following:

“Other analyses indicate that defective nuclear exosome function (*rrp6D*) also results in loss of H3K9me2 from centromeric repeats in the absence of RNAi (Reyes-Turcu et al., 2011). Cells lacking both Sir2 and Rrp6 have reduced H3K9me2, but in contrast to *sir2Δclr3D* it is not abolished (Figure S5E). This observation raises the possibility that Sir2 and Rrp6 act together to maintain H3K9me2 but requires further investigation to tease out their relationship”

- Please indicate the number of biological replicates performed to generate the data as well as information about the statistical test used to create the error bars to all Figure legends.

Response: We added this information to all figure legends.